# Early life stress causes sex-specific changes in adult fronto-limbic connectivity that differentially drive learning

Jordon D White[1], Tanzil M Arefin[2], Alexa Pugliese[1], Choong H Lee[2], Jeff Gassen[3], Jiangyang Zhang[2], Arie Kaffman[1]*

[1]Department of Psychiatry, Yale University School of Medicine, New Haven, United States; [2]Bernard and Irene Schwartz Center for Biomedical Imaging, Department of Radiology, New York University School of Medicine, New York, United States; [3]Department of Psychology, Texas Christian University, Fort Worth, United States

**Abstract** It is currently unclear whether early life stress (ELS) affects males and females differently. However, a growing body of work has shown that sex moderates responses to stress and injury, with important insights into sex-specific mechanisms provided by work in rodents. Unfortunately, most of the ELS studies in rodents were conducted only in males, a bias that is particularly notable in translational work that has used human imaging. Here we examine the effects of unpredictable postnatal stress (UPS), a mouse model of complex ELS, using high resolution diffusion magnetic resonance imaging. We show that UPS induces several neuroanatomical alterations that were seen in both sexes and resemble those reported in humans. In contrast, exposure to UPS induced fronto-limbic hyper-connectivity in males, but either no change or hypoconnectivity in females. Moderated-mediation analysis found that these sex-specific changes are likely to alter contextual freezing behavior in males but not in females.

*For correspondence:
arie.kaffman@yale.edu

## Introduction

Childhood maltreatment (CM) is a broad term used to define a heterogenous group of early adversities that range from severe bullying to physical, emotional, and/or sexual abuse (*White and Kaffman, 2019a*; *Teicher and Samson, 2016*). Exposure to CM increases the risk for the development of multiple psychopathologies and medical conditions over the lifespan (*Anda et al., 2006*; *Nemeroff, 2016*; *Teicher and Samson, 2016*; *Kaffman and Meaney, 2007*), with an estimated annual economic burden of $2 trillion in the United States alone (*Peterson et al., 2018*). While CM is recognized as a significant risk factor for abnormal brain development in industrialized countries, a thorough understanding of how CM impacts neurodevelopment and psychopathology in males and females is lacking (*White and Kaffman, 2019b*; *Bath, 2020*; *Bale and Epperson, 2015*; *Cameron et al., 2017*; *Gobinath et al., 2014*). Given that 30–40% of the adult population have experienced some form of CM (*Agorastos et al., 2019*), clarifying these issues is necessary in order to effectively diagnose and treat the enormous clinical and economic burden associated with CM (*White and Kaffman, 2019a*).

The study of CM-associated outcomes is generally framed from a cumulative risk perspective, where the risk of negative outcomes raises linearly with the number of maltreatment exposures (*Kessler et al., 1997*; *Anda et al., 2006*; *Chen et al., 2010*; *Evans et al., 2013*). Newer models have expanded this framework to highlight that different forms of CM fall along multiple dimensions, and this in turn more directly impacts later outcomes than number of CM instances per se (*McLaughlin et al., 2014*; *McLaughlin and Sheridan, 2016*). However, neither of these models has addressed the potential role that sex plays in moderating CM-associated outcomes (*Bath, 2020*;

*White and Kaffman, 2019b*). Further, much of the focus in the field to date has been on the role that sex plays in influencing the rate of psychopathology with little consensus and/or replication of findings (*White and Kaffman, 2019b*; *Cameron et al., 2017*; *Bale and Epperson, 2015*). Briefly, some studies report similar outcomes in males and females, some found females to be more sensitive to CM, while others see effects only in males (*White and Kaffman, 2019b*). A more nuanced approach suggests that sex-specific effects may depend on genetic vulnerability, the circuit involved, and the developmental stage when the stress occurred, or outcomes were assessed (*White and Kaffman, 2019b*; *Bath, 2020*; *Demaestri et al., 2020*).

Importantly, similar clinical presentation may be driven by different underlying mechanisms in males and females as reported for depression (*Labonté et al., 2017*) and neuropathic pain (*Sorge et al., 2015*). These findings suggest that sex-specific interventions might be needed to more effectively treat psychiatric and medical consequences of CM (*Bath, 2020*; *White and Kaffman, 2019b*). Promising progress on this issue comes from advanced imaging techniques (*Lerch et al., 2017*; *Le Bihan, 2003*; *Biswal, 2012*; *Moldrich et al., 2010*). The most reproducible findings are from structural MRI studies which show reduced hippocampal and corpus callosum volume in those exposed to CM, with a hint that these effects are larger in males compared to age-matched females (*Teicher and Samson, 2016*). A recent multi-center study using a large cohort of 3036 individuals did not find any significant changes in hippocampal size but noted reduced caudate size in females that was not seen in males exposed to CM (*Frodl et al., 2017*). Few studies have had sufficient power to directly test for CM by sex interactions on task mediated fMRI activation and connectivity using resting state fMRI (rsfMRI), or diffusion MRI (dMRI) (*White and Kaffman, 2019b*). Even within these studies, there is no agreement with regard to a specific pattern of task-mediated neuronal activation (*Crozier et al., 2014*; *Colich et al., 2017*; *Dannlowski et al., 2012*) or connectivity (*Herringa et al., 2013*; *Ohashi et al., 2019*).

This ambiguity highlights the difficulty of studying long-term consequences of complex adversity in human populations where the nature and severity of the CM may be different between and within studies and findings may be confounded with other psychological or medical co-morbidities; for a thorough review on this topic, see *Herringa, 2017*. Preclinical studies in rodents can bypass many obstacles faced in clinical research by precisely controlling the genetic composition of the animals, and the severity, duration, and complexity of the stress. Human imaging techniques, such as high-resolution MRI, dMRI, and rsfMRI, can be used in rodent models to characterize structural and functional outcomes that can be more directly compared to humans. Further, the contribution of structural and functional changes, measured via imaging, to behavioral alterations can be rigorously tested using optogenetics and chemogenetic tools (*Kaffman et al., 2019*). Although rodents and non-human primates exposed to early life stress (ELS) show similar behavioral and physiological outcomes reported in human studies, the vast majority of work has been done only in males (*White and Kaffman, 2019b*; *Bath, 2020*). Specifically, the few studies that used imaging to characterize outcomes in rodent models of ELS were done exclusively in males (*Bolton et al., 2018*; *Molet et al., 2016*; *Carlyle et al., 2012*; *Yan et al., 2017*; *Guadagno et al., 2018a*; *Sarabdjitsingh et al., 2017*; *Johnson et al., 2018*), with only one imaging study examining outcomes in both sexes (*Honeycutt et al., 2020*). The need to extend these findings to females is highlighted by a report of considerable neuroanatomical sex differences in the postnatal mouse brain (*Qiu et al., 2018*) and a recent study showing that maternal separation leads to sex-specific effects in amygdala (AMY) connectivity with the prefrontal cortex (PFC) in adolescent rats (*Honeycutt et al., 2020*).

Our lab has modified and built upon the widely used limited bedding and nesting (LBN) paradigm developed by Tallie Baram's lab (*Walker et al., 2017*) by extending the time frame dam and pups are exposed to LBN (from PND2-9 to PND0-25) and adding brief, 1 hr, bouts of maternal separation on an unpredictable schedule, that is, PND 14, 16, 17, 21, 22, and 25, followed by nest disruption. This ELS paradigm is termed unpredictable postnatal stress (UPS) and was designed to represent both a cumulative and multidimensional subtype of CM that allows us to test how complex adversity can alter neurodevelopment and behavioral outcomes in both male and female mice (*White and Kaffman, 2019b*). Balb/cByj mice were used due to their high sensitivity to stress (*Caldji et al., 2004*; *Zaharia et al., 1996*; *Francis et al., 2003*; *Carola et al., 2004*; *Tractenberg et al., 2016*; *Wei et al., 2010*; *Wei et al., 2012*; *McWhirt et al., 2019*; *Malki et al., 2015*; *Flint and Tinkle, 2001*; *Wei et al., 2014*; *Wei et al., 2015*; *Delpech et al., 2016*) and the ability of Balb/cByj dams to reliably maintain their litters in the complete absence of nesting material (*Wei et al., 2010*). Further,

we have previously shown that UPS produces an elevated anxiety phenotype in both adolescent and adult Balb/cByj offspring (*Johnson et al., 2018*). Using rsfMRI we found increased fronto-limbic connectivity in UPS male mice that included increased AMY–PFC and AMY–hippocampus connectivity, the strength of which was highly correlated with anxiety-like behaviors (*Johnson et al., 2018*). Interestingly, females did not show an elevated anxiety phenotype, suggesting that UPS may affect males and females differently. This original study did not include females in rs-fMRI analysis and is thus unable to speak to potential ELS by sex interactions on fronto-limbic connectivity patterns.

The current study aimed to replicate and extend our understanding of UPS-induced behavioral phenotypes and brain connectivity patterns in adult male and female mice. Our first objective was to replicate our previous findings that UPS increases anxiety-like behavior in male, but not female mice and to examine the effects of UPS, sex and their interaction on other exploratory behaviors (i.e. novel object) and contextual and cued fear conditioning. The second goal was to assess the effects of UPS and sex on local volumetric changes, microstructural alterations in white matter tracts, and structural connectivity using high resolution dMRI. Finally, we sought to replicate our fronto-limbic hyper connectivity rsfMRI findings in UPS male mice using dMRI tractography and to test whether similar changes would also be seen in female mice exposed to UPS.

## Results

### Effects of UPS on body weight and baseline stress levels

A main effect of rearing on body weight was consistently found at P14 ($F_{(1, 106)} = 36.66$, $p < 0.001$, $\eta_p^2 = 0.257$ *Figure 1A and B*) and P26 ($F_{(1, 104)} = 61.22$, $p < 0.001$, $\eta_p^2 = 0.303$, *Figure 1C*) showing that UPS animals were significantly smaller than control (CTL) animals. No significant effects of sex or an interaction were seen in 14- and 26-day-old pups ($p > 0.05$). The effect of rearing persisted into adulthood ($F_{(1, 28)} = 7.86$, $p = 0.009$ $\eta_p^2 = 0.219$, *Figure 1D*), with a significant main effect of sex ($F_{(1, 28)} = 80.31$, $p < 0.0001$, $\eta_p^2 = 0.741$), but no interaction between sex and rearing ($F_{(1, 28)} = 0.03$, $p = 0.87$, $\eta_p^2 = 0.001$). Serum from adult animals was processed for corticosterone levels. Results reveal a significant interaction of sex and rearing on corticosterone level ($F_{(1, 25)} = 9.37$, $p = 0.005$, $\eta_p^2 = 0.273$). Sidak's post-hoc analysis revealed a significant reduction in corticosterone levels in UPS compared to CTL males ($p = 0.01$, Cohn's d $= -1.4$), which was not seen in females ($p = 0.31$, Cohn's d $= 0.9$, *Figure 1E*). Further, there was a significant main effect of sex ($F_{(1, 25)} = 4.72$, $p = 0.04$, $\eta_p^2 = 0.159$), but no effect of rearing ($F_{(1, 25)} = 1.06$, $p = 0.31$, $\eta_p^2 = 0.041$; *Figure 1E*). Given the interaction between UPS and sex for corticosterone levels we also assessed the effects of UPS, sex, and the interaction on the normalized weight of the adrenal gland (*Figure 1F*). A significant effect of sex was also found for normalized adrenal gland weight ($F_{(1, 27)} = 117.10$, $p < 0.0001$, females > males, $\eta_p^2 = 0.813$), but with no significant effects for rearing ($F_{(1, 27)} = 0.60$, $p = 0.45$, $\eta_p^2 = 0.022$), or interaction ($F_{(1, 27)} = 0.05$, $p = 0.83$, $\eta_p^2 = 0.002$).

### Long-term consequences of UPS on behavior

Two-way ANOVA revealed no significant effects of sex, rearing, or an interaction on time spent in the center of the open field (sex: $F_{(1, 68)} = 1.34$, $p = 0.25$, $\eta_p^2 = 0.019$; rearing: $F_{(1, 68)} = 0.912$, $p = 0.34$, $\eta_p^2 = 0.013$; interaction: $F_{(1, 68)} = 3.37$, $p = 0.31$, $\eta_p^2 = 0.015$; *Figure 2A*), or on total distance traveled (sex: $F_{(1, 68)} = 0.908$, $p = 0.234$, $\eta_p^2 = 0.013$, rearing: $F_{(1, 68)} = 0.626$, $p = 0.43$, $\eta_p^2 = 0.009$; interaction: $F_{(1, 68)} = 1.959$, $p = 0.17$, $\eta_p^2 = 0.028$; *Figure 2B*). These behavioral outcomes were replicated in an additional cohort, see *Supplementary file 1*. Two-way ANOVA on time spent in the open arms of the EPM revealed a significant main effect of rearing ($F_{(1, 72)} = 7.411$, $p = 0.008$, $\eta_p^2 = 0.093$), where UPS-reared animals spent significantly more time in the open arms (*Figure 2C*). This effect was not replicated in a second cohort suggesting that UPS-induced elevations in exploration of open arms may not be robust. Further, there were no significant effects of sex ($F_{(1, 72)} = 0.054$, $p = 0.82$, $\eta_p^2 = 0.001$), or interaction ($F_{(1, 72)} = 0.49$, $p = 0.48$, $\eta_p^2 = 0.007$), and no differences were seen in closed arm exploration (sex: $F_{(1, 72)} = 0.35$, $p = 0.55$, $\eta_p^2 = 0.005$; rearing: $F_{(1, 72)} = 0.83$, $p = 0.37$, $\eta_p^2 = 0.011$; interaction: $F_{(1, 72)} = 2.25$, $p = 0.14$, $\eta_p^2 = 0.03$; *Figure 2D*). These effects were replicated in a second cohort, see *Supplementary file 1*.

UPS mice spent significantly less time exploring novel objects, rearing: $F_{(1, 70)} = 9.77$, $p = 0.003$, $\eta_p^2 = 0.123$ (*Figure 2E*), but with no significant interaction $F_{(1, 70)} = 0.306$, $p = 0.58$, $\eta_p^2 = 0.004$. These

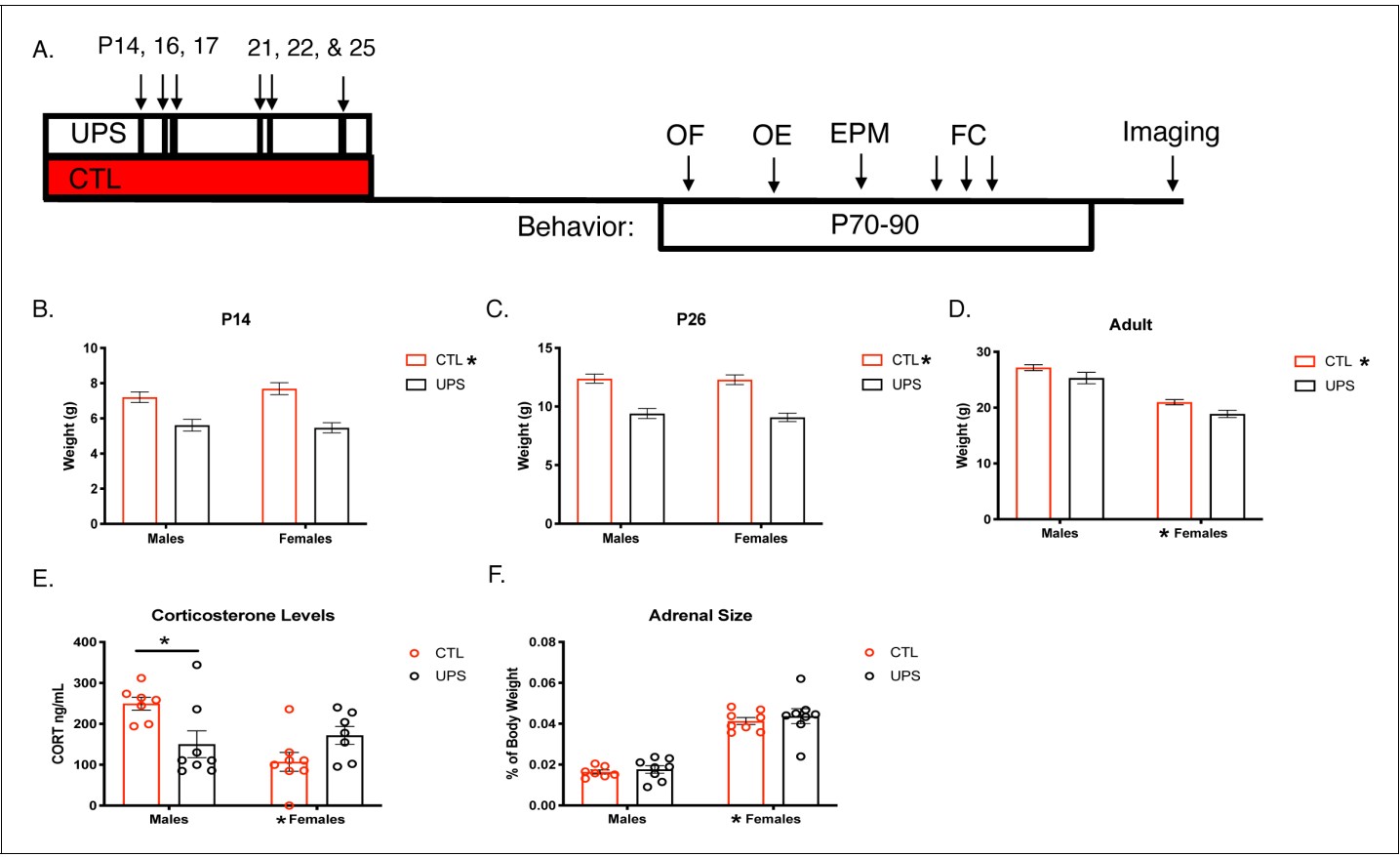

**Figure 1.** Long-lasting effects of UPS on body weight and stress response. (**A**) Timeline. (**B–D**) Body weights across the lifespan (P14 and 26: n = 23–33, adulthood: n = 8 for each rearing and sex group, see *Figure 1—source data 1* for raw data). (**E**) Baseline corticosterone levels in adulthood (n = 7–8 per rearing and sex group). (**F**) Adult adrenal size normalized to body weight (n = 7–8 per rearing and sex group). UPS: unpredictable postnatal stress, CTL: control, OF: open field, OE: object exploration, EPM: elevated plus maze, FC: fear conditioning. Mean and SEM, *p<0.05. The online version of this article includes the following source data for figure 1:

**Source data 1.** Raw data for body weight, corticosterone level, and adrenal weight.

effects were seen in an additional cohort (*Supplementary file 1*). In cohort 1, there was also a significant effect of sex: $F_{(1, 70)}=22.53$, p<0.001, $\eta_p^2 = 0.243$, but this finding was not replicated in the additional cohort (*Supplementary file 1*).

Learned freezing behavior was then assessed in a contextual and cued-fear conditioning paradigm. Initial freezing prior to the onset of shocks revealed no inherent differences in baseline activity between sexes or rearing conditions (sex: $F_{(1, 68)}=0.28$, p=0.60, $\eta_p^2 = 0.004$; rearing: $F_{(1, 68)}=2.34$, p=0.13, $\eta_p^2 = 0.033$; interaction: $F_{(1, 68)}=0.02$, p=0.89, $\eta_p^2 < 0.001$), findings that were all replicated in an additional cohort (sex: $F_{(1, 62)}=0.22$, p=0.64, $\eta_p^2 = 0.004$; rearing: $F_{(1, 62)}=0.004$, p=0.95, $\eta_p^2 < 0.001$; interaction: $F_{(1, 62)}=0.717$, p=0.40, $\eta_p^2 = 0.011$). Repeated measures ANOVA revealed a significant effect of freezing behavior over time (Greenhouse–Geisser: $F_{(2.2686, 272)}=91.38$, p<0.001, $\eta_p^2 = 0.573$; cohort 2: $F_{(2.464, 248)}=70.50$, p<0.001, $\eta_p^2 = 0.532$) where animals froze more as the number of shocks experienced increased regardless of sex or rearing condition. No other significant within-subject effects were noted (p>0.1). After 24 hr, animals were placed back into the training context. Two-way ANOVA revealed a significant main effect of rearing ($F_{(1, 68)}=18.52$, p<0.001, $\eta_p^2 = 0.214$; *Figure 2F*) where UPS-reared animals froze significantly less to the original training context compared to CTL-reared counterparts, an effect that was replicated in cohort 2 (*Supplementary file 1*). No significant effects of sex ($F_{(1, 68)}=1.88$, p=0.18, $\eta_p^2 = 0.027$) or an interaction ($F_{(1, 68)}=3.37$, p=0.07, $\eta_p^2 = 0.047$) were seen in cohort 1 or cohort 2 (*Supplementary file 1*). Together, these findings suggest that the UPS rearing condition leads to reproducible deficits in context encoding or retrieval.

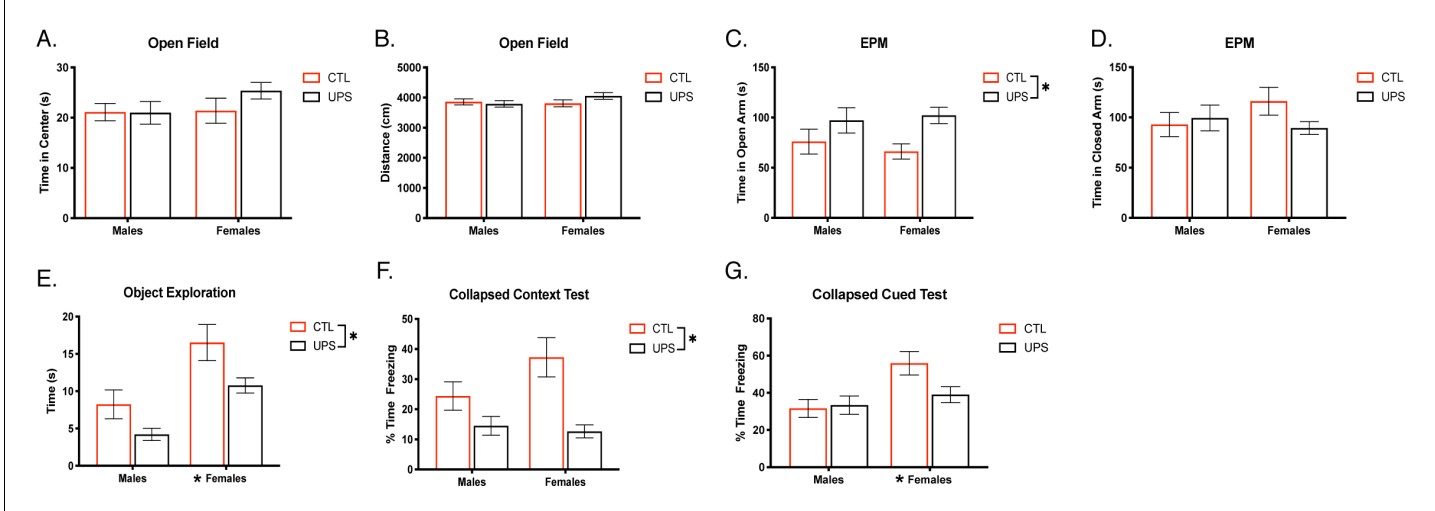

**Figure 2.** Effects of unpredictable postnatal stress and sex on adult behavior. Time in the center (**A**) and distance traveled in the OF (**B**). Time spent in the open (**C**) and closed arms (**D**) of the EPM. Time exploring objects in an arena (**E**). Average freezing behavior in contextual fear conditioning test (**F**) and during cue presentations in novel context (**G**) (n = 12–23 per rearing and sex group). Mean and SEM, *p<0.05. *Figure 2—source data 1*.
The online version of this article includes the following source data and figure supplement(s) for figure 2:

**Source data 1.** Raw data for behavioral outcomes.
**Figure supplement 1.** Exposure to acoustic-cue in a novel environment led to robust increase in freezing but no consistent effects of rearing, sex, or interaction.
**Figure supplement 2.** Behavioral testing in a third cohort of adult mice found consistent effects of rearing in the object exploration and contextual fear conditioning, but not acoustic-mediated fear learning.

Mice showed minimal freezing to the novel context on day 3, but tone exposure produced significantly higher freezing behavior indicating appropriate cue-mediated learning (*Figure 2—figure supplement 1*). There were no consistent effects of rearing, sex, or interaction on baseline freezing and responses to tone-on and tone-off episodes (*Figure 2—figure supplement 1*). When collapsing tone-on episodes, two-way ANOVA for cue-recall revealed a significant main effect of sex (F (1, 68) =8.57, p=0.005, $\eta_p^2$ = 0.112, *Figure 2G*), with females freezing more than males to the cue, an effect replicated in cohort 2 (*Supplementary file 1*). Further, there were no significant effects of rearing condition or an interaction on cue-induced freezing (rearing: (F (1, 68)=2.204), p=0.14, $\eta_p^2$ = 0.031; interaction: (F (1, 68)=3.32), p=0.07, $\eta_p^2$ = 0.047; *Figure 2G*), with similar outcomes seen in the second (*Supplementary file 1*).

To further demonstrate the robustness of the rearing effect on outcomes in the novel exploration test (*Figure 2E*) and contextual fear condition (*Figure 2F*) we replicated these findings in additional cohorts of mice (*Figure 2—figure supplement 2*). Moreover, using linear mixed-effect analyses we show that behavioral outcomes in these two tests are not likely to be mediated by a litter effect (*Supplementary file 2*).

## Voxel-based volumetric changes

Next, we obtained high resolution dMRI scans (n = 6 mice per rearing condition and sex, for a total of 24 adult mice) and conducted a 2 × 2 whole-brain voxel-based morphometric analysis to identify local volumetric changes affected by UPS, sex, and their interaction. Using a minimal voxel cluster of 25 and a false discovery rate of 0.1 (p<0.011), we identified several volumetric changes that were affected by condition regardless of sex (*Figure 3*). Volumetric changes were relatively small (20–30%) and included increased volumes in the nucleus accumbens, olfactory bulb (OB), cingulate cortex, subregions within the sensory cortex, ventral hippocampus (vHC), and AMY in UPS mice. UPS mice showed reduced volumes in the frontal cortex, PFC, subregions of the motor cortex (MO), the sensory cortex, fimbria, striatum, and thalamus when compared to CTL mice (*Supplementary file 3*). Relatively few areas showed volumetric changes between males and females and their identification required less stringent false discovery rate of 0.3 (p<0.0107) and minimum cluster size at 25. Using

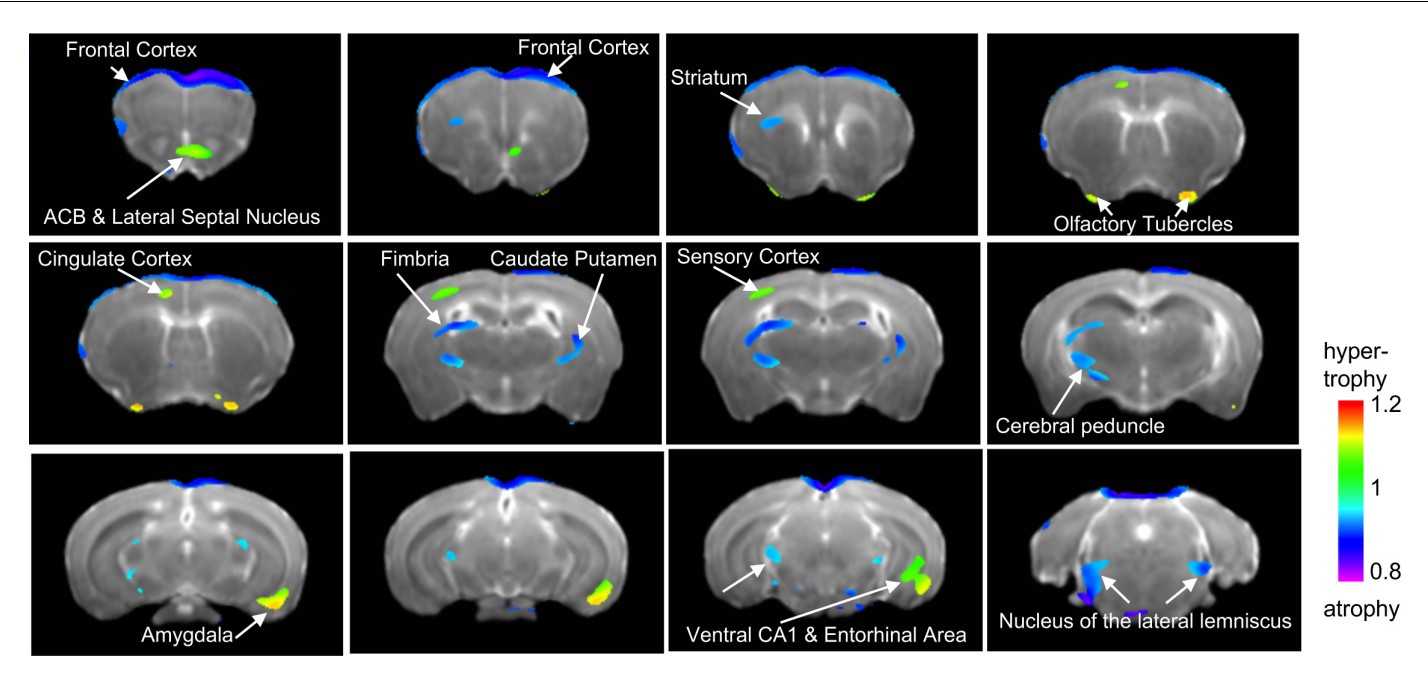

**Figure 3.** Main effect of unpredictable postnatal stress (UPS) on local volumetric changes. Minimal cluster size >25 voxels, FDR < 0.1, p<0.0105. 1 = no change; <1 (blue) reduced volume in UPS; >1 (green) increased in UPS (n = 6 per rearing and sex group). Nucleus accumbens – ACB. *Figure 3—source data 1*.

The online version of this article includes the following source data and figure supplement(s) for figure 3:

**Source data 1.** Matlab codes used to conduct the 2 × 2 analyses for *Figures 3* and *4*.

**Figure supplement 1.** Effect of sex on local volumetric changes using voxel-based morphometric analysis, FDR < 0.3, p<0.0107, minimal cluster size = 25 voxels.

**Figure supplement 2.** Areas that show potentially significant interactions between unpredictable postnatal stress and sex for volumetric changes using voxel-based morphometric analysis (p<0.05 uncorrected, minimal cluster size = 20 voxels).

these parameters, males showed increased volumes in subregions of the BNST, medial preoptic area, OB, and dorsal hippocampus (dHC) and reduced volume in parts of the MO, sensory cortex, striatum, and thalamus when compared to the female mice (*Figure 3—figure supplement 1*). No brain regions showed UPS by sex interaction after correcting for multiple comparisons and adjusting the minimal cluster size to more than 20. Therefore, an exploratory analysis was conducted using uncorrected analysis (p<0.05 uncorrected, minimal cluster size >20) and the results are shown in *Figure 3—figure supplement 2*.

## Fractional anisotropy (FA)

A 2 × 2 ANOVA revealed significant effect of UPS on FA in numerous brain regions (minimum cluster size >25, FDR = 0.1, p<0.007) with moderate effect size of 40–50% (*Figure 4*). Increased FA in UPS was most pronounced in subregions of the sensory cortex, hypothalamus, and AMY with reduced FA was seen in the rostral corpus callosum and stria medularis (*Figure 4*). Similar to the volumetric analysis, the effects of sex on FA were subtle requiring lowering the FDR to 0.3 (*Figure 4—figure supplement 1*) and an exploratory uncorrected characterization of the interaction between UPS and sex interaction is available in *Figure 4—figure supplement 2*.

## Tractography

We previously found increased rsfMRI connectivity between the AMY and the PFC and between the AMY and the hippocampus (both ventral and dorsal) in UPS male mice (*Johnson et al., 2018*) and wanted to test whether a similar increase in structural connectivity is seen using tractography. Moreover, our volumetric and FA analyses have revealed long-term structural changes associated with UPS, with only subtle effects of sex, and little evidence to support interaction between UPS and sex.

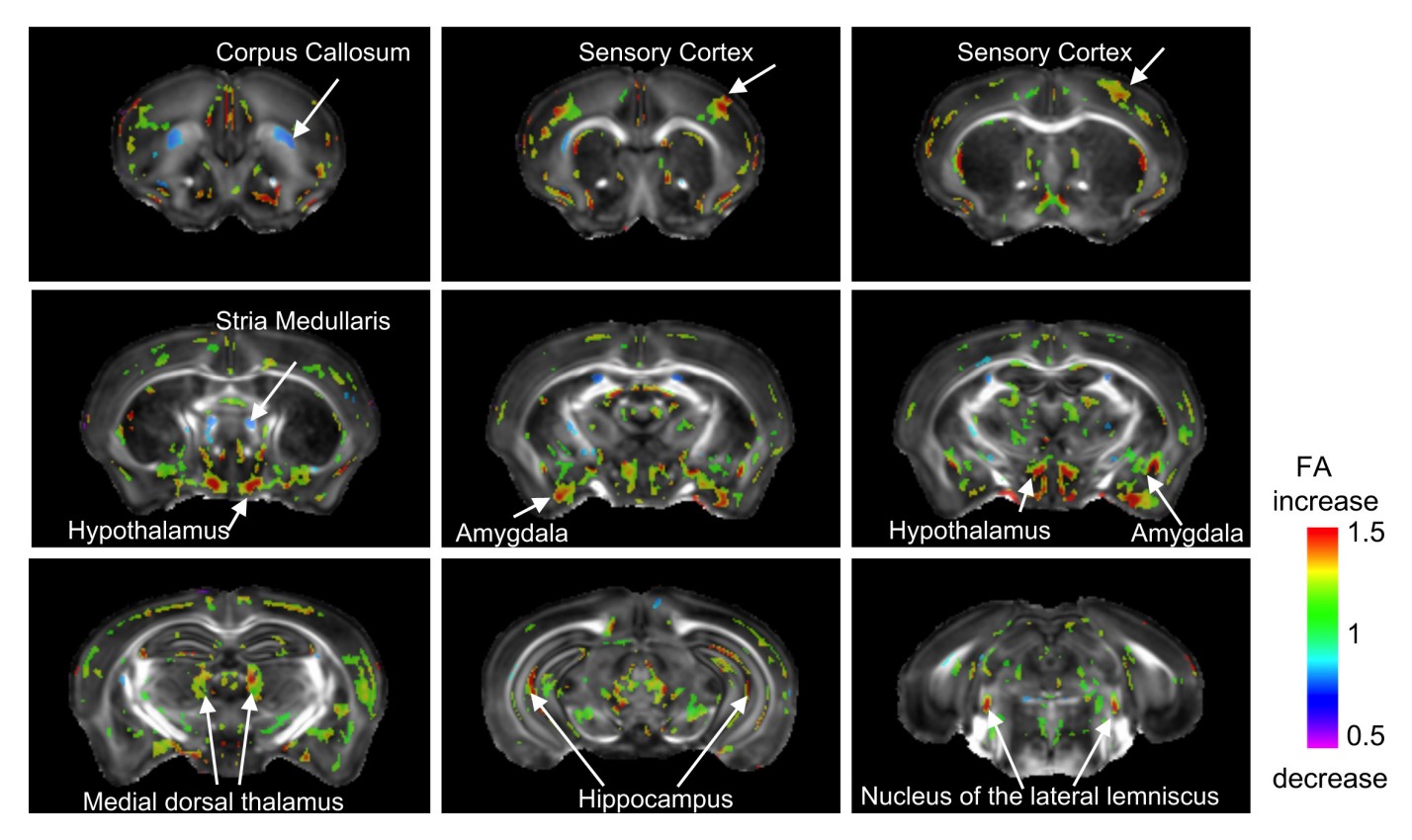

**Figure 4.** Main effect of unpredictable postnatal stress (UPS) on local changes in FA. Minimal cluster size >25 voxels, FDR < 0.1, p<0.007. 1 = no change; <1 (blue) reduced volume in UPS; >1 (green/red) increased in UPS (n = 6 per rearing and sex group). *Figure 3—source data 1*. The online version of this article includes the following figure supplement(s) for figure 4:

**Figure supplement 1.** Effect of sex on local changes in fractional anisotropy (FA), FDR < 0.3, p<0.0004, minimal cluster size = 1 voxel.
**Figure supplement 2.** Areas that show potentially significant interactions between unpredictable postnatal stress and sex for fractional anisotropy (FA) (p<0.05 uncorrected, minimal cluster size = 20 voxels).

Thus, it became important to test whether a similar pattern would be seen using tractography. To address this issue, we first confirmed that there is a good agreement between our dMRI projections and the expected anterograde connectivity available from the Allen Mouse Brain Connectivity Atlas (*Figure 5—figure supplement 1*). Next, we quantified ipsilateral streamlines between the AMY, hippocampus (vHC and dHC), and the PFC in both the right and left hemispheres. Repeated measures ANOVA was first conducted to ascertain whether laterality existed in the effect of UPS, sex, and their interaction on connectivity. In most cases there was no significant effect of laterality allowing us to combine streamlines data from the left and right hemispheres; a detailed analysis for each hemisphere is available in *Figure 5—figure supplements 2* and *3*.

## AMY–PFC

No significant effects were found within subjects (p>0.05), but a 2 × 2 ANOVA found significant main effects of sex (F (1, 20)=8.90, p=0.007, $\eta_p^2$ = 0.292), rearing condition (F (1, 20)=6.48, p=0.02, $\eta_p^2$ = 0.289), and interaction (F (1, 20)=19.269, p<0.001, $\eta_p^2$ = 0.483) on the number of streamlines between the AMY and the PFC (*Figure 5A*). Sidak's post-hoc analysis showed a significant effect of sex within the UPS condition (p<0.001, Cohn's d = −3.5), but not the CTL condition (p=0.33, Cohn's d = 0.5). Further, there was a significant effect of rearing condition for males (p<0.001, Cohn's d = 2.9), but not for females (p=0.21, Cohn's d = −0.6). Overall, AMY–PFC connectivity was significantly increased in UPS-reared males but was not altered in UPS-reared females (*Figure 5A*).

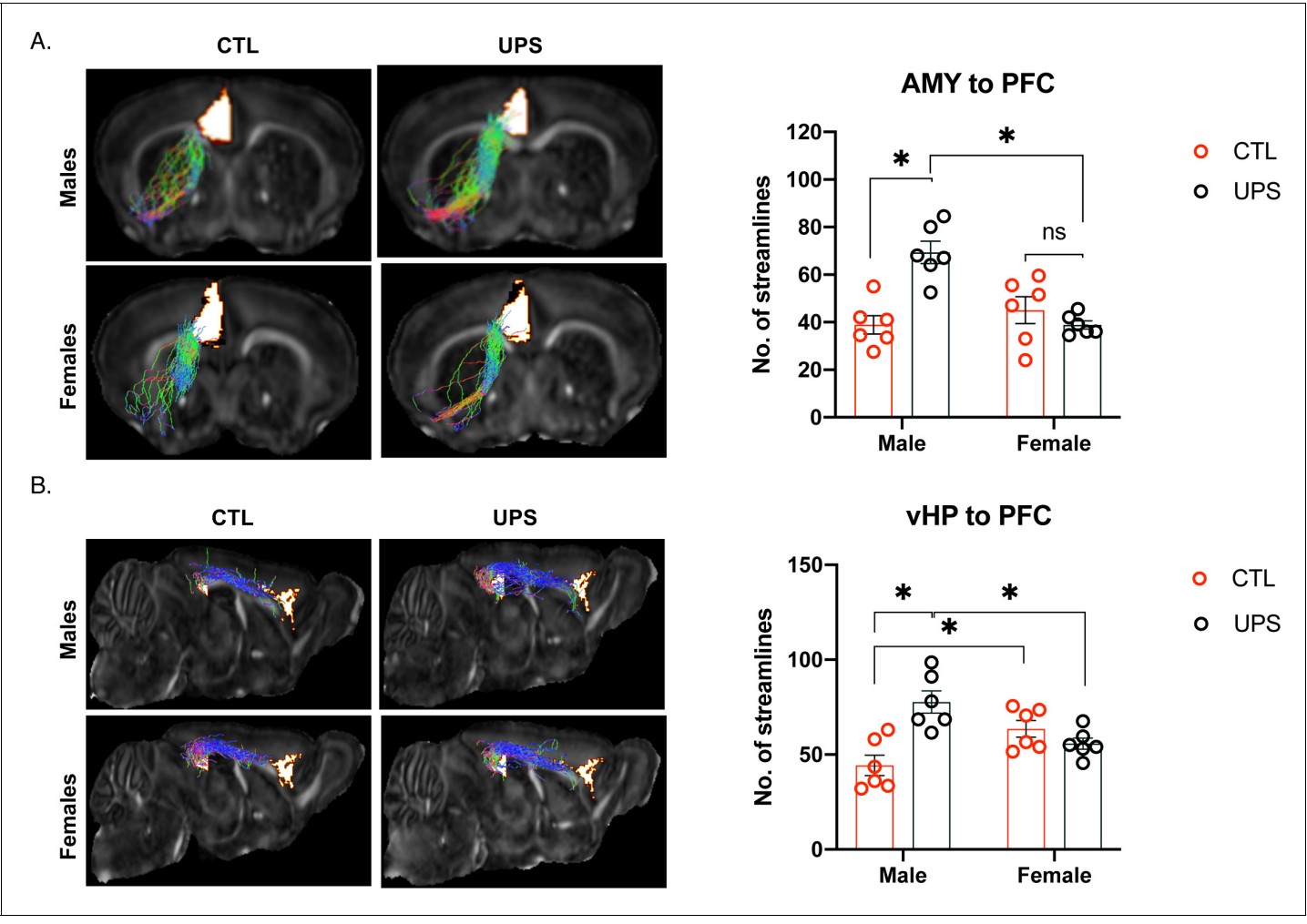

**Figure 5.** Unpredictable postnatal stress differentially alters PFC connections with the AMY and vHC in males and females. (**A**) Representative images and quantification of AMY–PFC tractography. (**B**) Representative images and quantification of vHC–PFC tractography. AMY – amygdala, PFC – prefrontal cortex, vHP – ventral hippocampus (n = 6 per rearing and sex group). Mean and SEM. *p<0.05.
The online version of this article includes the following figure supplement(s) for figure 5:

**Figure supplement 1.** dMRI tractography shows good agreement with the anterograde tracing reported by the Allen Mouse Brain Connectivity Atlas (AMBCA) (*Oh et al., 2014*).
**Figure supplement 2.** Amygdala–prefrontal cortex tractography for the left (**A** and **C**) and the right (**B** and **D**) hemispheres.
**Figure supplement 3.** Ventral hippocampus–prefrontal cortex tractography for the left (**A** and **C**) and the right (**B** and **D**) hemispheres.

## vHC–PFC

No significant effects were found within subjects (p>0.05), and there was no significant effect of sex (F $(1, 20)$=0.08, p=0.78, $\eta_p^2$ = 0.004). A significant effect of rearing (F $(1, 20)$=7.07, p=0.02, $\eta_p^2$ = 0.261) and interaction between rearing and sex (F $(1, 20)$=18.416, p<0.001, $\eta_p^2$ = 0.479) were found for the number of streamlines between the vHC and the PFC (*Figure 5B*). Sidak's post-hoc analysis showed a significant effect of sex within the UPS condition (p=0.004, Cohn's d = −1.9) and the CTL condition (p=0.01, Cohn's d = 1.6) whereby within the CTL condition females have increased number of streamlines over males, but this pattern was reversed in the UPS condition. Further, there was a significant effect of rearing condition for males (p<0.001, Cohn's d = 2.4), but not females (p=0.26, Cohn's d = −0.9). Overall, vHC–PFC showed a similar pattern of connectivity to that seen between the AMY and the PFC, with UPS increasing the number of streamline projections in males but not females (*Figure 5B*).

### AMY–hippocampus

Repeated measures ANOVA revealed a significant within-subject main effect of hemisphere on the number of streamlines between the AMY complex and the hippocampal formation (*Figure 6A*). Specifically, the left hemisphere showed significantly higher connectivity compared to the right hemisphere (Greenhouse–Geisser $F_{(1, 20)}=8.183$, $p<0.01$, $\eta_p^2 = 0.29$; *Figure 6—figure supplement 1A and B*). Further, there was a significant interaction between sex and hemisphere (Greenhouse–Geisser $F_{(1, 20)}=17.82$, $p<0.01$, $\eta_p^2 = 0.471$) that when unpacked showed a significant effect of sex within the right hemisphere ($p=0.001$, Cohn's d = 1.3) but not within the left ($p=0.29$, Cohn's d = −0.4), and a significant effect of hemisphere for females ($p<0.001$, Cohn's d = 1.8), but not males ($p=0.35$, Cohn's d = −0.2), *Figure 6—figure supplement 1A and B*. Together, these findings suggest a small reduction in right hemisphere connectivity in females (*Figure 6—figure supplement 1B*). No other significant effects were found within subjects ($p>0.05$). Between-subject analysis found a significant interaction between sex and rearing condition ($F_{(1, 20)}=18.74$, $p<0.001$, $\eta_p^2 = 0.484$)

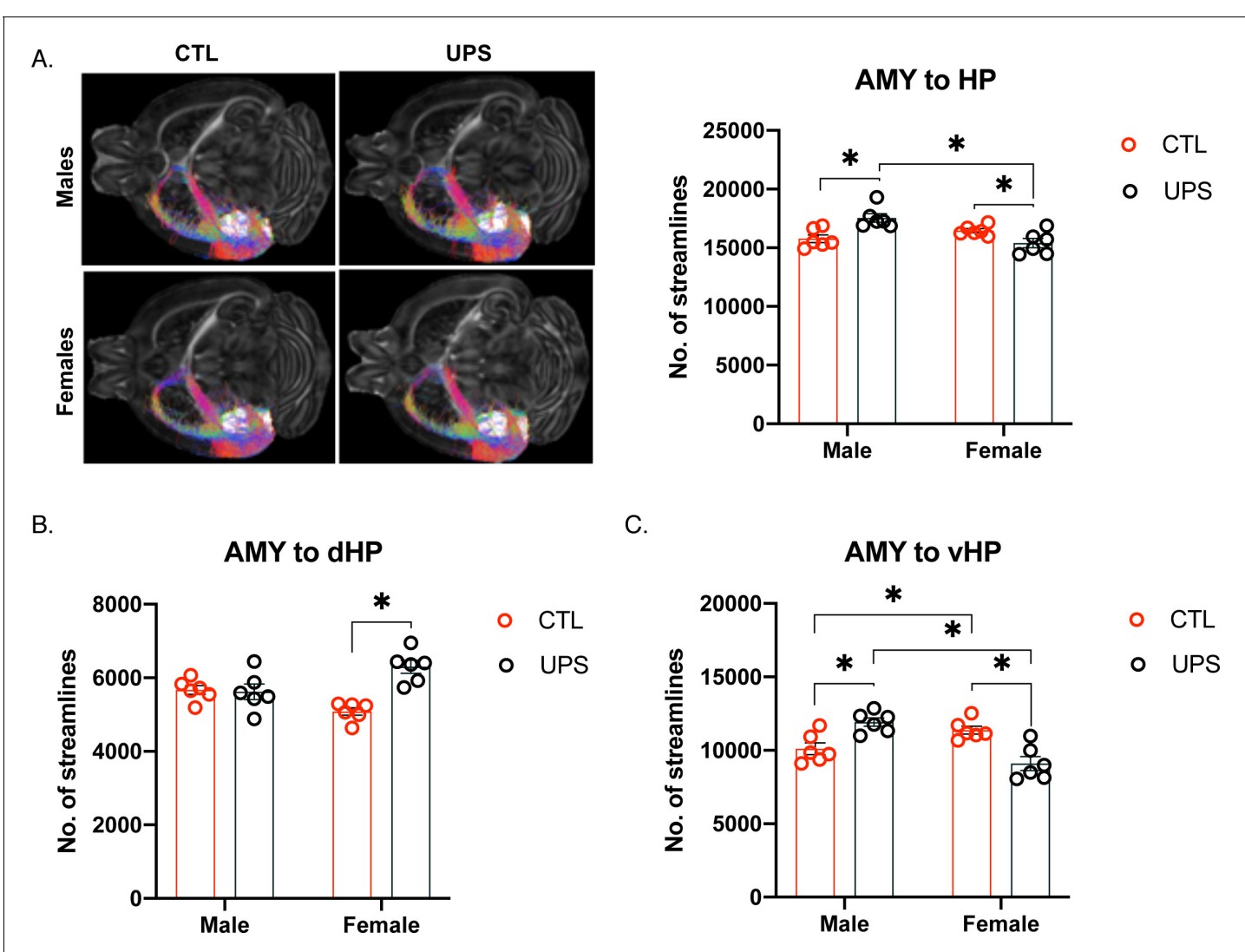

**Figure 6.** Unpredictable postnatal stress causes sex-specific alterations in AMY–hippocampus connections. (**A**) Representative images and quantification of the number of streamline connections between AMY and the entire hippocampus. Number of streamline connections between the AMY and the dorsal hippocampus (**B**) and between the AMY and the vHP (**C**). AMY – amygdala, dHP – dorsal hippocampus, vHP – ventral hippocampus (n = 6 per rearing and sex group). Mean and SEM, *p<0.05.

The online version of this article includes the following figure supplement(s) for figure 6:

**Figure supplement 1.** Amygdala (AMY)–hippocampus tractography for the left (**A, C, E, and F**) and the right hemispheres (**B, D, G, and H**).

that when unpacked revealed a significant effect of sex within the UPS condition (p<0.001, Cohn's d = 2.3), but not within the CTL condition (p=0.16, Cohn's d = −1.1) where UPS females show a reduced number of streamlines compared to UPS males. Further, a significant effect of rearing condition was found within males (p=0.001, Cohn's d = 2.1) where UPS males showed increased connectivity compared to CTL males, but the pattern was significantly reversed in females (p=0.03, Cohn's d = −1.4; *Figure 6A*). To further explore this, streamlines were specifically quantified between the AMY and dHC versus vHC.

## AMY–dHC

No significant effects were found within subjects (p>0.05, *Figure 6—figure supplement 1E and G*), but significant between-subject effects were seen (*Figure 6B*). There was no significant effect of sex (F (1, 20)=0.093, p=0.76, $\eta_p^2$ = 0.005), but a significant effect of rearing condition (F (1, 20)=13.62, p=0.001, $\eta_p^2$ = 0.405) and interaction (F (1, 20)=5.96, p=0.001, $\eta_p^2$ = 0.444). Unpacking the interaction revealed a significant increase in connectivity in females (p<0.001; UPS > CTL, Cohn's d = 3.5) but not males (p=0.83; Cohn's d = −0.1 *Figure 6B*).

## AMY–vHC

Repeated measures ANOVA revealed a significant within-subject main effect of hemisphere on the number of streamlines with higher connectivity in the left compared to the right hemisphere (Greenhouse–Geisser F (1, 20)=5.318, p=0.03, $\eta_p^2$ = 0.21). There was a significant interaction of sex and hemisphere (Greenhouse–Geisser F (1, 20)=6.93, p=0.02, $\eta_p^2$ = 0.257) that when unpacked revealed a significant effect of sex on the right hemisphere (p=0.005, Cohn's d = 1.0) but not left (p=0.86, Cohn's d = −0.1). Further, females show decreased connectivity in the right hemisphere (p=0.002, Cohn's d = −1.0) compared to the left, a pattern not observed in males (p=0.82, Cohn's d = 0.1), *Figure 6—figure supplement 1F and H*. No other significant effects were found within subjects (p>0.05), but significant between-subject effects were seen. Results reveal an insignificant main effect of rearing condition (F (1, 20)=0.386, p=0.54, $\eta_p^2$ = 0.019), but a significant effect of sex (F (1, 20)=4.473, p=0.047, $\eta_p^2$ = 0.183) and interaction (F (1, 20)=30.89, p<0.001, $\eta_p^2$ = 0.607 *Figure 6C*). Sidak's post-hoc analysis found enhanced connectivity in UPS compared to CTL males (p=0.002, Cohn's d = 2.1), but reduced connectivity in UPS females compared to CTL females (p<0.001, Cohn's d = −2.4). These suggest that the alterations in connectivity seen in UPS between the AMY to the whole hippocampal formation (*Figure 6A*) are mainly driven by changes in AMY connectivity with the vHC (*Figure 6C*).

## AMY–vHC tractography and freezing behavior

Given the large number of streamline projections between the AMY and vHC (*Figure 6C*) and their role in footshock consolidation (*Huff et al., 2016*) and contextual fear learning and retrieval (*Fanselow and Dong, 2010*; *Zhu et al., 2014*; *Rudy and Matus-Amat, 2005*), we conducted a moderated-mediation analysis to examine the role that sex plays in impacting the effects of UPS on the number of AMY–vHC streamlines in the left hemisphere (*Figure 7A*, path A) and its relationship to freezing behavior in contextual fear conditioning (*Figure 7A*, path B). This analysis revealed that both rearing condition (b = 1822.67, SE = 654.59, t = 2.78, p=0.01, 95% CI [457.14, 3188.20]), and sex (b = 2063.33, SE = 654.59, t = 3.15, p=0.005, 95% CI [697.80, 3428.86]) significantly predicted the number of streamlines between the AMY and vHC (*Figure 7A*, path A). Consistent with our previous analysis, however, there was a significant interaction between rearing and sex (b = −3966.33, SE = 925.73, t = −4.29, p=0.0004, 95% CI [−5897.48, −2035.18]). Specifically, for males, the number of AMY–vHC streamlines was greater for UPS animals compared to CTL animals (b = 1822.67, SE = 654.59, t = 2.78, p=0.01, 95% CI [457.14, 3188.20]) while the opposite effect was found for females (b = −2143.67, SE = 654.59, t = −3.28, p=0.004, 95% CI [−3509.20, −778.14]). While both the main effects of number of AMY–vHC streamlines (b = −0.0003, SE = 0.0001, t = −3.61, p=0.002, 95% CI [−0.0005, −0.0001]) and sex (b = 0.34, SE = 0.13, t = 2.63, p=0.02, 95% CI [0.07, 0.62]) significantly predicted time spent freezing, there was also a significant interaction between sex and rearing condition (b = 0.0003, SE = 0.0001, t = 2.75, p=0.01, 95% CI [0.0001, 0.0006], *Figure 7A*, path B). For males, a greater number of AMY–vHC streamlines predicted decreased time spent freezing (b = −0.0003, SE = 0.0001, t = −3.61, p=0.002, 95% CI [−0.0005, −

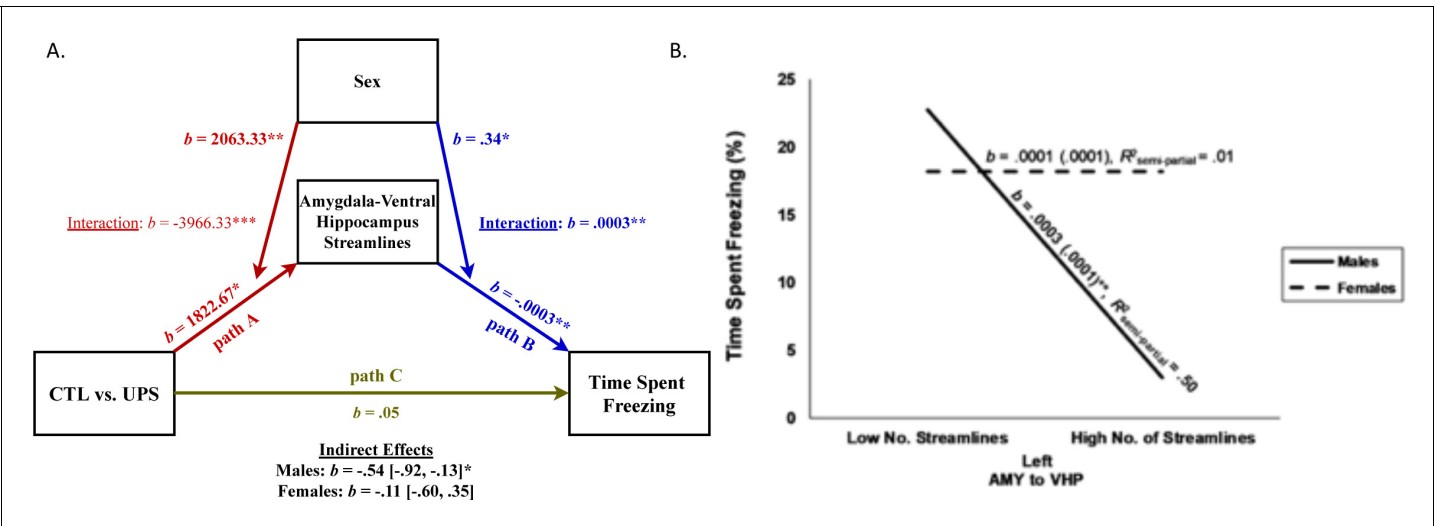

**Figure 7.** Moderated-mediation model of freezing behavior. (**A**) Path model testing the effects of sex on number of amygdala (AMY)–ventral hippocampus (vHP) streamlines (left hemisphere) in mice exposed to unpredictable postnatal stress (UPS; path A) and the moderating effects of sex on the relationship between number of streamlines connections and freezing behavior (path B). Direct effect of UPS on freezing behavior after controlling for the indirect effects of streamline number and its interaction with sex is shown as path C, and the overall significance for the indirect moderating paths is shown for males and females at the bottom. (**B**) Interaction between number of streamlines (left hemisphere) between AMY and vHP and sex on freezing behavior. High and low number of streamlines correspond to one standard deviation above and below the mean of this variable, respectively. Overall, streamlines predict freezing behavior in males but not females. ***p≤0.001, **p≤0.01, *p≤0.05.

The online version of this article includes the following source data for figure 7:

**Source data 1.** SPSS syntax used for the moderator analysis.

0.0001]), but there was no significant relationship between number of AMY–vHC streamlines and freezing behavior for females (b = 0.0001, SE = 0.0001, t = 0.70, p=0.49, 95% CI [−0.0001, 0.0002]). In contrast to the findings in the left hemisphere, the number of AMY–vHC streamlines in the right hemisphere did not significantly predict freezing behavior for either male or female animals (p>0.5).

Overall, tests of statistical mediation revealed a significant indirect effect of rearing condition on freezing behavior via number of left AMY–vHC streamlines for males (b = −0.54, SE = 0.20, 95% CI [−0.92, −0.13]), but not females (b = −0.11, SE = 0.21, 95% CI [−0.60, 0.35], *Figure 7A*, paths A and B, and bottom of figure for summary stats). The direct effect of rearing condition on freezing behavior (path C) was no longer significant after controlling for the mediating influence of streamline number and its interaction with sex (b = 0.05, SE = 0.18, t = 0.27, p=0.79, 95% CI [−0.33, 0.43]).

Together, these results support the notion that reduced freezing behavior seen in UPS males is driven, at least in part, by increased connectivity between the AMY and vHC in the left hemisphere. For males, UPS rearing (compared to CTL rearing) led to an increased number of AMY–vHC streamlines in the left hemisphere, which in turn led to reduced freezing behavior. The direct effect of rearing condition on time spent freezing became nonsignificant when the number of streamline connections was controlled by the model (*Figure 7C*, path C). This indicates that UPS-mediated changes in connectivity represent a major contributor to freezing behavior in male mice. This was not the case for females where exposure to UPS led to a decrease in the number of AMY–vHC streamlines in the left hemisphere, a reduction that did not significantly predict freezing behavior (*Figure 7B*).

## Global and regional connectivity

A recent dMRI study in humans found similar global connectivity changes in adult males and females exposed to CM (*Ohashi et al., 2019*). To test whether global connectivity is differentially affected in male and female UPS mice, we generated a 14 × 14 matrix (*Figure 8A*). This network included 182 tractograms in each hemisphere that were used to calculate global efficiency and small-worldness using Graph Theory (*Bullmore and Sporns, 2009*). No significant effects of hemisphere were found within subjects (p>0.05) allowing us to test the effects of rearing, sex and their interaction across

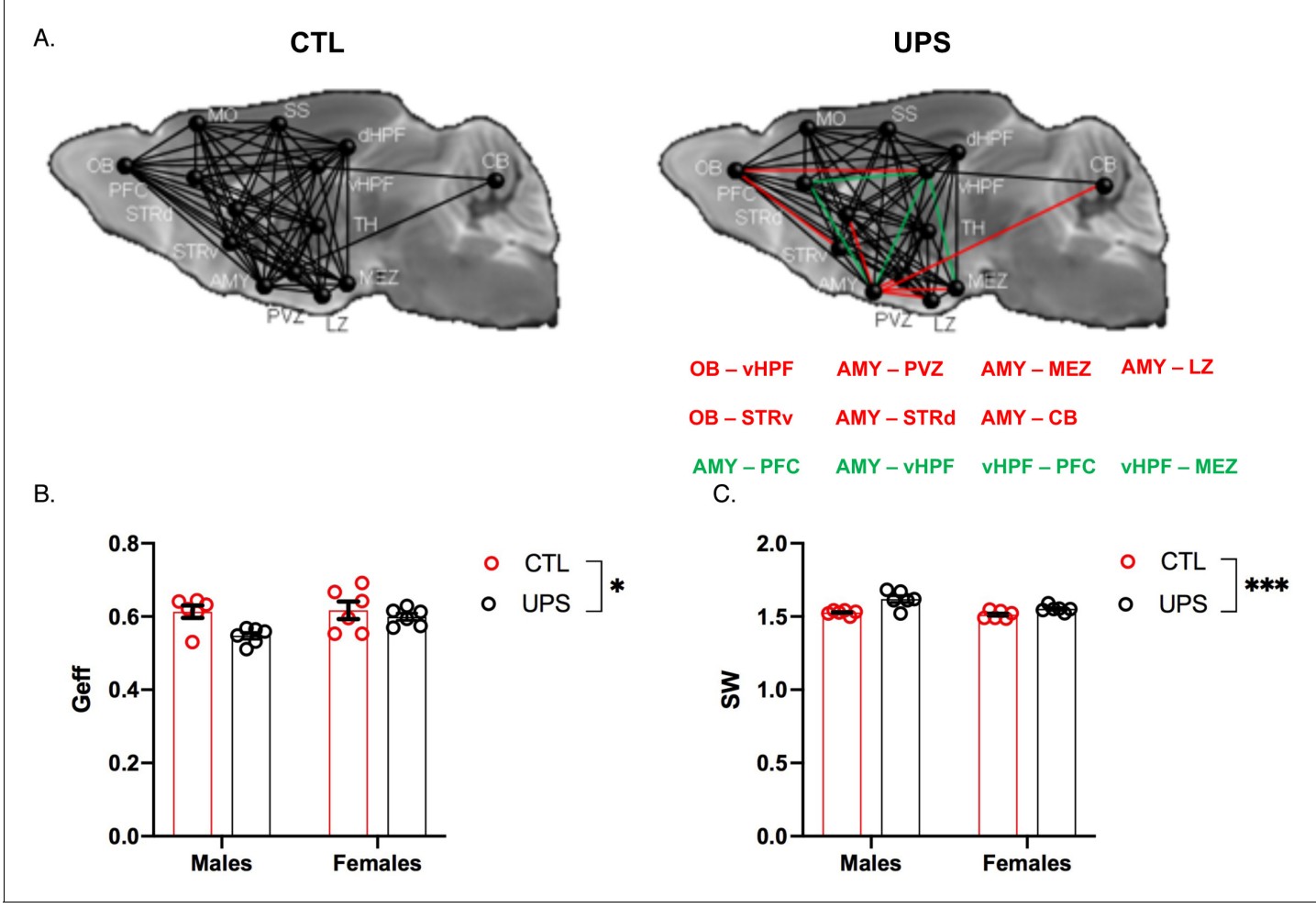

**Figure 8.** Effects of unpredictable postnatal stress (UPS) and sex on global connectivity. (**A**) Schematic representation of global connectivity for control (CTL) and UPS. Red lines indicate reduced connectivity and green lines indicate increased connectivity compared to CTL group. (**B**) Quantification of global efficiency (Geff) and (**C**) Small-worldness (SW). (n = 6 per rearing and sex group). Mean and SEM, *p<0.05. Abbreviations: amygdala (AMY), cerebellum (CB), dorsal hippocampus (dHP), dorsal striatum (STRd), lateral hypothalamic zone (LZ), medial hypothalamic zone (MEZ), motor cortex (MO), olfactory bulb (OB), periventricular zone (PVZ), prefrontal cortex (PFC), somatosensory cortex (SS), thalamus (TH), ventral hippocampus (vHP), ventral striatum (STRv).

The online version of this article includes the following source data for figure 8:

**Source data 1.** GRETNA codes used for global connectivity analysis.

both hemispheres. A 2 × 2 ANOVA found a main effect of rearing on global network efficiency (F (1, 20)=6.66, p=0.018, $\eta_p^2$ = 0.25) with no significant effects of sex (F (1, 20)=2.94, p=0.10, $\eta_p^2$ = 0.13), or interaction (F (1, 20)=2.21, p=0.15, $\eta_p^2$ = 0.10, *Figure 8B*). There were also significant effects of rearing (F (1, 20)=21.77, p<0.0005, $\eta_p^2$ = 0.52) and sex (F (1, 20)=8.56, p=0.0084, $\eta_p^2$ = 0.3) on small-worldness (*Figure 8C*), but no significant interaction (F (1, 20)=3.50, p=0.076, $\eta_p^2$ = 0.15). Overall, exposure to UPS reduced global network efficiency and increased small-worldness in both males and females. These findings are consistent with human imaging data (*Ohashi et al., 2019*) and are mainly driven by reduced connectivity between the AMY and to a lesser extent the OB, with several other brain regions including the hypothalamus, striatum, hippocampus, and cerebellum (CB) (*Figure 8A*, red lines).

Previous work in humans has shown that alterations in AMY's centrality can distinguish between symptomatic and asymptomatic individuals exposed to CM (*Ohashi et al., 2019*). To test this issue in UPS mice we examined the effects of UPS, sex, and their interaction on nodal clustering, efficiency, and degree of centrality for the right and left AMY (*Figure 9A*). No significant differences on

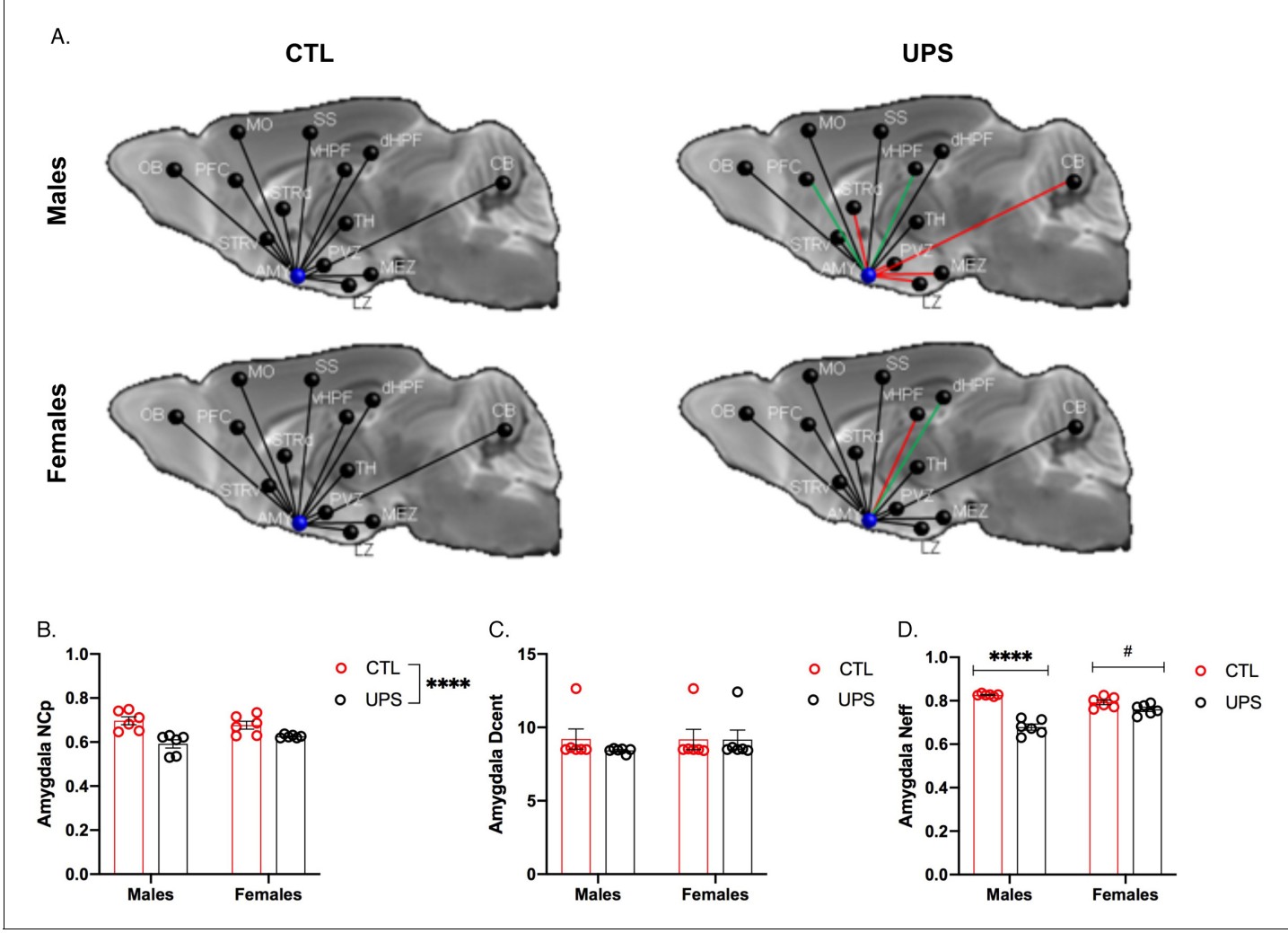

**Figure 9.** Effects of unpredictable postnatal stress and sex on AMY regional connectivity. (A) Schematic representation of AMY connectivity with other brain regions. Red lines indicate reduced connectivity and green lines indicate enhanced connectivity compared to same sex group. (B) Quantification for nodal clustering coefficient (NCp), (C) degree of centrality (Dcent), and (D) nodal efficiency (Neff). Abbreviations: amygdala (AMY), cerebellum (CB), dorsal hippocampus (dHP), dorsal striatum (STRd), lateral hypothalamic zone (LZ), medial hypothalamic zone (MEZ), motor cortex (MO), olfactory bulb (OB), periventricular zone (PVZ), prefrontal cortex (PFC), somatosensory cortex (SS), thalamus (TH), ventral hippocampus (vHP), ventral striatum (STRv). Mean and SEM, *p<0.05.

The online version of this article includes the following source data for figure 9:

**Source data 1.** GRETNA codes used to assess amygdala connectivity.

these outcomes were found between the left and the right AMY allowing us to collapse across hemi-spheres. A 2 × 2 ANOVA found a significant effect of rearing on the nodal clustering coefficient (F (1, 20)=24.08, p<0.0001, $\eta_p^2$ = 0.55), where UPS-reared animals displayed a smaller nodal clustering coefficient compared to CTL-reared animals (*Figure 9B*). No significant effects of sex or interaction were found for degree of centrality (*Figure 9B*). A large effect size that was highly significant was seen for the interaction between rearing and sex for nodal efficiency (F (1, 20)=32.38, p<0.0001, $\eta_p^2$ = 0.62; *Figure 9C*). Follow-up Sidak's post-hoc analysis revealed a highly significant reduction in AMY nodal efficiency in UPS males compared to CTL-reared males (p<0.0001, Cohn's d = 6.0), a dif-ference that was much smaller and was not significant in females (p=0.07, Cohn's d = 1.3). Further inspection of the connectivity maps clarified that reduced nodal AMY efficiency seen in male UPS mice is driven by reduced connectivity between the AMY and structures such as striatum,

hypothalamus, and CB (*Figure 9A*, red lines). Together, these findings highlight the exquisite specificity by which UPS alters structural connectivity in male and female mice.

## Generalizability of the sex-specific tractography findings

To further test the generalizability of the sex-specific tractography findings, we increased the number of mice scanned to 12–13 per rearing and sex, for a total of 50 mice. This larger cohort of mice confirmed sex-specific changes in AMY–PFC, vHC–PFC, AMY–hippocampus, and AMY–vHC connectivity. Connectivity between the AMY and the dHC showed a significant rearing effect, but no significant rearing by sex interaction (*Figure 10*). Measurements of global connectivity such as global efficiency and small-worldness revealed highly significant effects of rearing consistent with the findings seen in the smaller cohort of mice (*Figure 11*). All the 13 previously identified connectomes affected in the subgroup of animals (*Figure 8*) were also differentially regulated in the larger cohort of mice. Two additional connectomes, not identified previously (AMY–sensory cortex and AMY–MO), showed reduced connectivity in UPS in this larger group of mice (*Figure 11*). Outcomes of AMY regional connectivity were also very similar to those reported in our initial group of mice (*Figure 12*). Together, these findings demonstrate robust sex-specific changes in fronto-limbic connectivity in adult mice exposed to UPS.

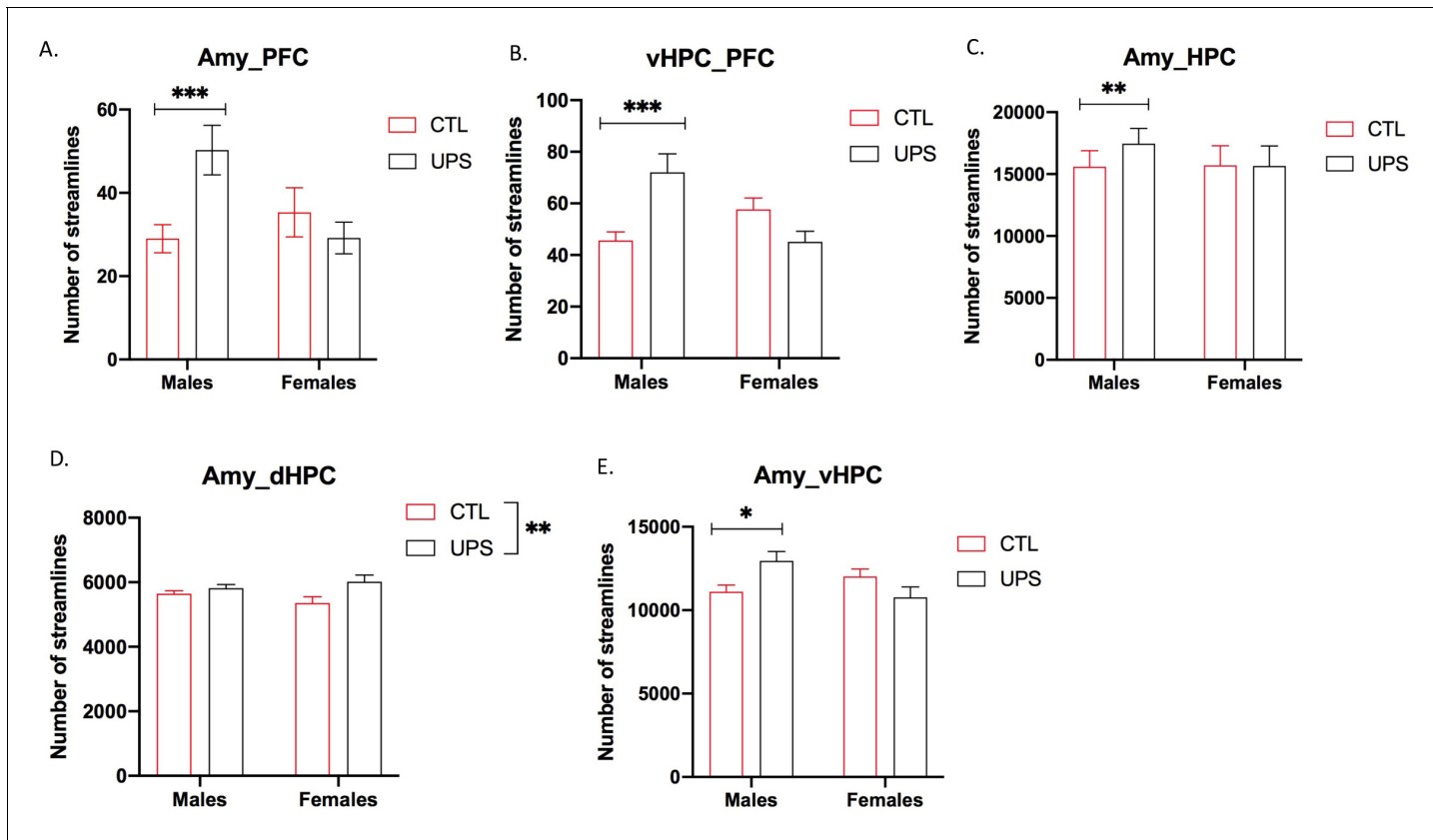

**Figure 10.** Fronto-limbic connectivity using an expanded cohort of animals (n = 12–13 per rearing and sex, total n = 50 mice). (**A**) Amygdala (AMY)–prefrontal cortex (PFC) connectivity revealed rearing by sex interaction (F (1, 46)=7.82, p=0.0075) that was driven by increased connectivity in unpredictable postnatal stress (UPS) males (p=0.0061) but not UPS females (p=0.63). (**B**) vHC–PFC showed rearing by sex interaction (F (1, 46)=14.99, p=0.0003) that was due to increased connectivity in UPS males (p=0.0009) but not UPS females (p=0.17). (**C**) There was a significant rearing by sex interaction (F (1, 46)=5.57, p=0.023) for AMY–PFC connectivity that was driven by sex-specific increase in connectivity in UPS males (p=0.0034) but not UPS females (p=0.99). (**D**) AMY–dHPC showed significant rearing effect (F (1, 46)=7.33, p=0.0095), with no significant effects of sex or interaction. (**E**) AMY–vHPC connectivity revealed a significant rearing by sex interaction (F (1, 46)=8.98, p=0.0044) that was due to increased connectivity in UPS males (p=0.026) but not UPS females (p=0.19). Graphs represent mean and SEM, *p<0.05, **p<0.01, ***p<0.005.

The online version of this article includes the following source data for figure 10:

**Source data 1.** Raw data for fronto-limbic connectivity in the extended cohort.

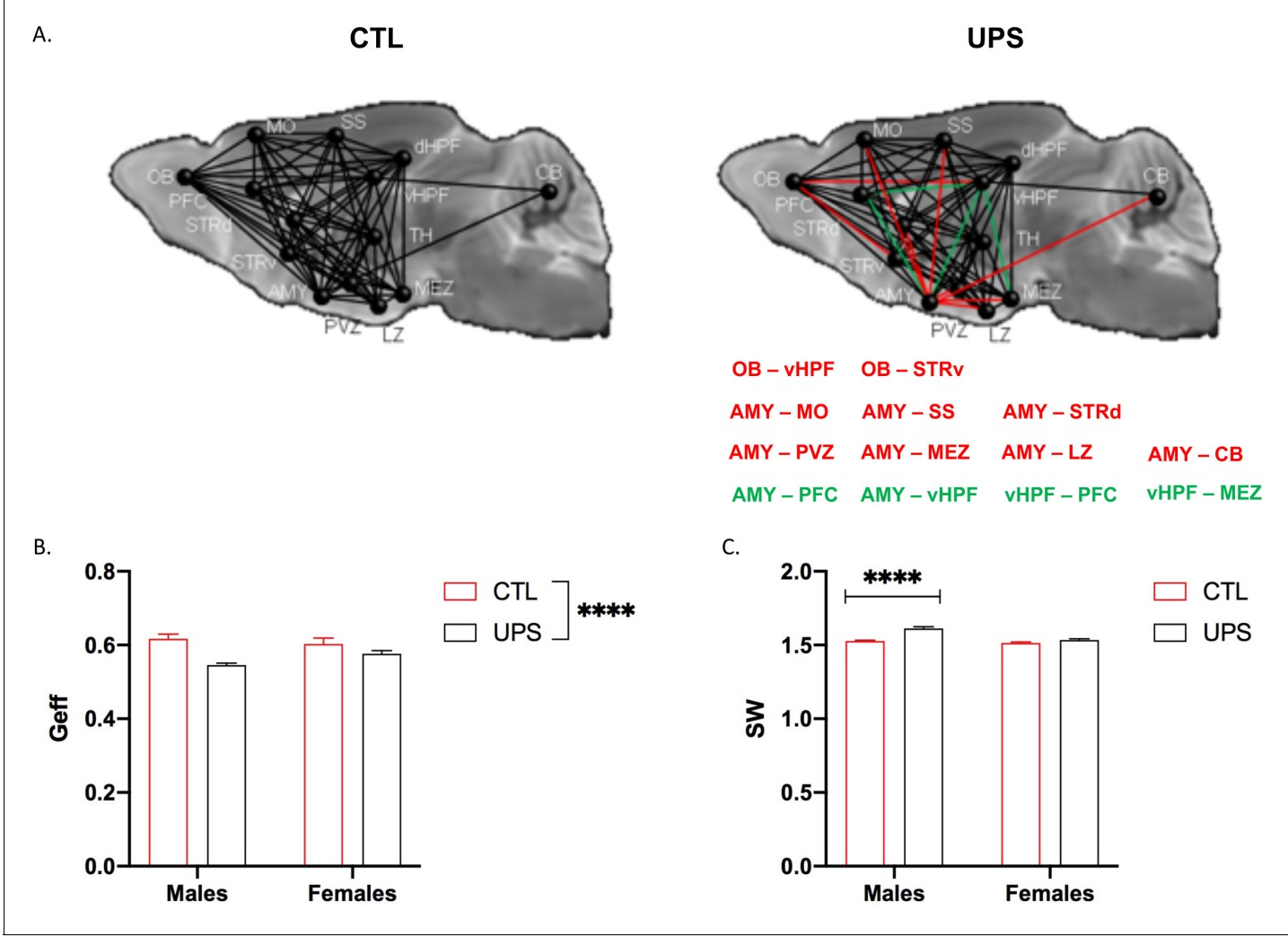

**Figure 11.** Global connectivity using an expanded cohort of animals (n = 12–13 per rearing and sex, total n = 49 mice). (**A**) Schematic representation of global connectivity for control (CTL) and unpredictable postnatal stress. Red lines indicate reduced connectivity and green lines indicate increased connectivity compared to CTL group. (**B**) Global efficiency (Geff) showed a significant effect of rearing (F (1, 45)=18.68, p<0.0001), no significant effects of sex (F (1, 45)=0.53, p=0.47), and a trend for interaction (F (1, 45)=3.92, p=0.054). (**C**) Small-worldness (SW) showed significant effects of rearing (F (1, 45)=38.38, p<0.0001), sex (F (1, 45)=29.6, p<0.0001), and interaction (F (1, 45)=14.13, p=0.0005), with Sidak's post-hoc analysis showing significant increase in males (p<0.0001) but not females (p=0.18). Mean and SEM, ****p<0.00001. Abbreviations: amygdala (AMY), cerebellum (CB), dorsal hippocampus (dHP), dorsal striatum (STRd), lateral hypothalamic zone (LZ), medial hypothalamic zone (MEZ), motor (MO), olfactory bulb (OB), periventricular zone (PVZ), prefrontal cortex (PFC), somatosensory cortex (SS), thalamus (TH), ventral hippocampus (vHP), ventral striatum (STRv). The online version of this article includes the following source data for figure 11:

**Source data 1.** Raw data for *Figures 11* and *12*.

## Discussion

The present study demonstrates that UPS, a complex model of ELS in the mouse, produces long-lasting behavioral and central anatomical changes, some of which are moderated by sex. The role that sex plays in CM-associated outcomes is a growing area of interest (*White and Kaffman, 2019b*; *Bath, 2020*), and it is becoming increasingly clear that the influence of sex may only be seen in certain behaviors, neurodevelopmental processes, and/or specific circuits (*White and Kaffman, 2019b*; *Bath, 2020*; *Demaestri et al., 2020*). This is in line with the findings reported here. For example, here we report similar impact of UPS on body weight, object exploration, contextual conditioning, and global connectivity in males and females. Male and female UPS mice also showed similar changes in local volumetric and FA, but our sample size for these measures (n = 6) was not

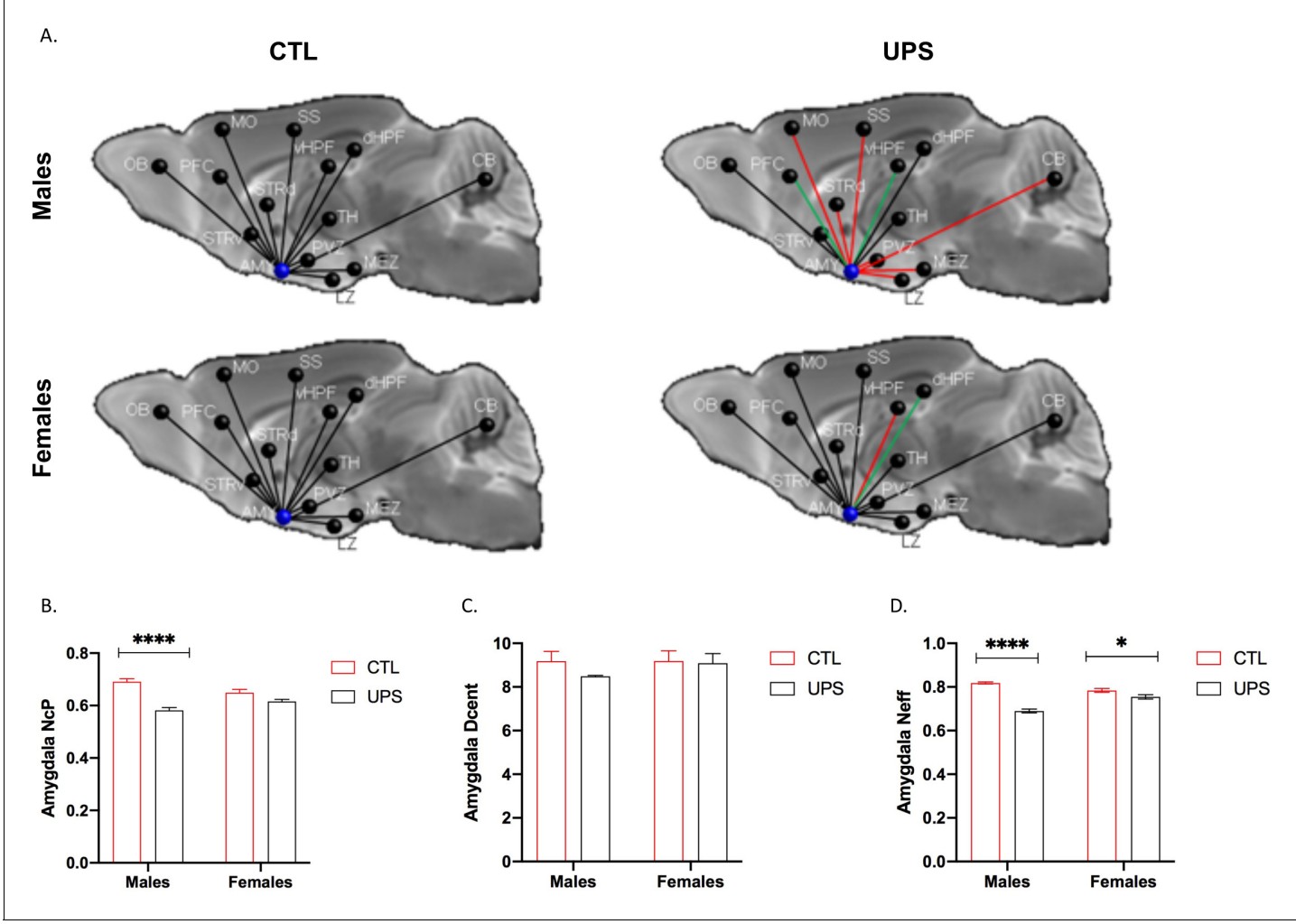

**Figure 12.** AMY regional connectivity using an expanded cohort of animals (n = 12–13 per rearing and sex, total n = 49 mice). (**A**) Schematic representation of AMY connectivity with other brain regions. Red lines indicate reduced connectivity and green lines indicate enhanced connectivity compared to same sex group. (**B**) Nodal centrality (NcP) showed significant rearing (F (1, 45)=42.56, p<0.0001) and interaction (F (1, 45)=12.23, p=0.0011), but no significant effect of sex (F (1, 45)=0.15, p=0.70). Sidak's post-hoc analysis showing significant increase in nodal centrality in males (p<0.0001) but not in females (p=0.078). (**C**) Degree of centrality (Dcent) revealed no significant effects of rearing, sex, or interaction. (**D**) Nodal efficiency showed significant rearing (F (1, 45)=89.41, p<0.0001) and interaction (F (1, 45)=35.40, p<0.0001), but no significant effect of sex (F (1, 45)=3.39, p=0.072). Sidak's post-hoc analysis found significant increase in nodal centrality in males (p<0.0001) and in females (p=0.036). Abbreviations: amygdala (AMY), cerebellum (CB), dorsal hippocampus (dHP), dorsal striatum (STRd), lateral hypothalamic zone (LZ), medial hypothalamic zone (MEZ), motor (MO), olfactory bulb (OB), periventricular zone (PVZ), prefrontal cortex (PFC), somatosensory cortex (SS), thalamus (TH), ventral hippocampus (vHP), ventral striatum (STRv). Mean and SEM, *p<0.05, ****p<0.0001. *Figure 11—source data 1*.

sufficiently powered to detect significant sex by rearing interactions after correcting for multiple testing. Nevertheless, this sample size revealed robust moderating effects of sex on UPS-induced outcomes in fronto-limbic connectivity and AMY nodal efficiency, findings that were further confirmed using a larger cohort of mice.

To the best of our knowledge this is the first paper to utilize high resolution dMRI to assess the effects of ELS on brain structure in both males and females. Using these methods, we demonstrated that males and females exposed to UPS showed several changes that are consistent with those previously reported in human and animal literature. Specifically, UPS leads to reduced global network efficiency and increased small-worldness that resemble those seen in a large cohort of individuals exposed to CM (*Ohashi et al., 2019*). Global efficiency, the ability of a network to propagate parallel information, increases during brain development and is associated with improved cognition (*Bassett and Bullmore, 2017*; *Farahani et al., 2019*; *Huang et al., 2015*; *Bullmore and Sporns,*

*2009*). The reduced global efficiency seen in UPS mice suggests abnormal maturation of this feature and is consistent with the cognitive deficits seen in the same animals. Small-worldness, a highly conserved network pattern across species, decreases during development (*Bassett and Bullmore, 2017*; *Bullmore and Sporns, 2009*; *Lupfer et al., 2013*). Thus, the increased small-worldness seen in UPS animals is further support of a premature network.

In terms of volumetric changes, UPS mice showed several outcomes that were also reported in humans (summarized in *Supplementary file 3*). These include reduced volumes in the frontal cortex (*Carrion et al., 2001*; *De Bellis et al., 2002*; *Heim et al., 2013*), reduced corpus callosum volume (*Teicher and Samson, 2016*), increased anterior cingulate (*Zuo et al., 2019*), and increased AMY volume (*Pechtel et al., 2014*; *Mehta et al., 2009*; *Tottenham et al., 2010*). It is important to note, however, that increased anterior cingulate and AMY volumes have not been consistently reported in humans (*Teicher and Samson, 2016*; *Frodl et al., 2017*), but increased AMY volumes are seen in non-human primates (*Coplan et al., 2014*) and juvenile rats exposed to LBN (*Guadagno et al., 2018b*). With regard to FA changes, UPS mice showed reduced FA in the corpus callosum which is one of the most robust findings in the human literature (*Teicher and Samson, 2016*; *McCarthy-Jones et al., 2018*). In contrast, the increased vHC volume seen in UPS mice is diametrically opposite to a large body of work showing reduced hippocampal volume in individuals exposed to CM (*Baker et al., 2013*; *Cohen et al., 2006*; *Teicher and Samson, 2016*). However, a recent analysis using over 3000 subjects found no statistical differences in hippocampal volume (*Frodl et al., 2017*), underscoring the challenges involved in conducting this work in humans. Inconsistent outcomes have also been reported in rats exposed to LBN, with one report indicating increased volume (*Guadagno et al., 2018a*) and a second publication noting reduced hippocampal volume (*Molet et al., 2016*). Additional work is therefore needed to replicate our findings, clarify mechanisms, and the contribution of this expansion in vHC to connectivity and behavioral outcomes.

Our sample size was not sufficiently powered to identify sexually dimorphic changes after rigorous correction for multiple comparisons, and the relatively lower resolution of dMRI compared to T1/T2-weighted MRI and differences in tissue contrasts may have also limited our ability to capture these differences. Nevertheless, we have identified several of the previously known sexually dimorphic brain regions including the BNST and medial preoptic area in the hypothalamus, hippocampus, thalamus, OB, striatum, and the sensory cortex (*Figure 3—figure supplement 1*). These findings are consistent with a previous study in C57BL/6 mice (*Qiu et al., 2018*) and in humans (*Gillies and McArthur, 2010*; *Ruigrok et al., 2014*; *Goldstein et al., 2001*), underscoring the conserved nature of these sexual dimorphic alterations.

Our dMRI tractography shows good agreement with anterograde viral tracing reported by the Allen Mouse Brain Connectivity Atlas (*Figure 5—figure supplement 1*), suggesting that the tractography data shown here represents true anatomical projections. Using this approach we found a robust increase in fronto-limbic connectivity in males exposed to UPS (*Figures 5* and *6*). These structural changes are consistent with our previous resting-state fMRI findings in UPS males (*Johnson et al., 2018*) and those reported in male rats exposed to LBN (*Bolton et al., 2018*). Similar patterns were seen in both the left and the right hemispheres (*Figure 5—figure supplements 1* and *2* and *Figure 6—figure supplement 1*) and were further confirmed using a larger cohort of mice (*Figure 10*). The observation that the hyper connectivity is unique to males and is not seen in females reveals a novel sex-dependent effect of UPS on this specific circuit. Sex-specific alterations in connectivity have been reported in adolescents and young adults exposed to ELS (*Crozier et al., 2014*; *Herringa et al., 2013*; *Colich et al., 2017*) but not in adult subjects (*Ohashi et al., 2019*; *Dannlowski et al., 2012*). Nevertheless, increased AMY connectivity was associated with higher levels of symptomatology including anxiety and depression in adulthood (*Ohashi et al., 2019*), suggesting that findings in UPS males may have translational utility for some types of ELS in humans. Exposure to maternal separation also induced sex-specific changes in the connectivity between the basolateral AMY and the PFC in juvenile rats; however, these changes appear to be more prominent in females (*Honeycutt et al., 2020*). Together, these findings highlight intriguing sex-dependent effects of varying ELS paradigms on fronto-limbic connectivity in rodents.

Despite clear differences in UPS-induced fronto-limbic connectivity between males and females, their behavioral profiles were remarkably similar and raise the possibility that alterations in connectivity may lead to different behavioral outcomes in males and females. To address this issue, we conducted a moderated-mediation analysis to test whether sex was moderating the relationships

between UPS-induced changes in connectivity between the AMY and the vHC and their resulting influence on freezing behavior. We focused on these connections for several reasons, that is, UPS induced opposing outcomes in males and females (increased in males and reduction in females, *Figure 6D*), there were clear volumetric changes in both areas, and these areas play an important role in fear learning. Our analysis suggests that alterations in connectivity in males, but not females, is in part responsible for the reduced freezing behavior seen in UPS mice (*Figure 7*). Future work should use optogenetic or chemogenetic approaches to further test the conclusions drawn from this moderated-mediation model, but the findings reported here are consistent with other examples showing that similar behavioral outcomes can be driven by different mechanisms in males and females (*Labonté et al., 2017*; *Sorge et al., 2015*).

Previous work has shown that ELS alters brain laterality (*Denenberg et al., 1980*; *Sherman et al., 1980*; *Zhang et al., 2005*; *Guadagno et al., 2018a*; *Sullivan and Gratton, 2002*), but the exact mechanisms responsible for these changes are yet to be elucidated. Most of our findings were seen on both the left and the right hemispheres. Nevertheless, we report three interesting findings, all of which are related to laterality within the AMY–vHC connections. First, there was a significant increase in connectivity on the left compared to the right regardless of sex. Second, there was a significant effect of sex on the right hemisphere with females showing reduced connections compared to males. Third, moderated-mediation analysis found higher number of projections between the AMY and vHC on the left, but not the right, predicting male freezing behavior in the contextual fear conditioning.

It is important to note that we were unable to replicate our UPS-induced anxiety-like behavior in the open field or the EPM (*Johnson et al., 2018*). One possible explanation for this failure to replicate may be due to differences in the implementation of the paradigm. For instance, the current studies were conducted in a room that was roughly 10 times quieter than the room utilized in previous work (see Animals heading within the Materials and methods section) and the work was conducted by different individuals. These outcomes also highlight the sensitivity of the OF and the EPM tests to multiple variables such as sex of the experimenter, phase of the light/dark cycle, degree of illumination in the arena, noise, smells, and order of behavioral testing (*Tractenberg et al., 2016*; *Alves et al., 2019*; *Murthy and Gould, 2018*; *Lehmann and Feldon, 2000*). Our inability to replicate behavioral outcomes in the EPM in the first and second cohorts underscores the challenges of using this test even under practically identical conditions (*Supplementary file 1*) and highlights the need to develop more robust behavioral tests of anxiety; see *Oberrauch et al., 2019*; *Meyer et al., 2019*; *Prevot et al., 2019* for some encouraging examples. Reduced baseline corticosterone levels were seen in males, but not female UPS mice (*Figure 1E*). This outcome is consistent with a recent meta-analysis showing an overall blunted corticosterone levels in individuals exposed to early adversity (*Bunea et al., 2017*) and might be due to sex-specific changes in connectivity between the AMY and the hypothalamus (*Figure 9A*).

Additional work is also needed to characterize maternal care in UPS and its potential for inducing sex-specific changes; see *Bath, 2020* for a comprehensive review on this topic. Consistent with our previous work (*Johnson et al., 2018*) we found that UPS caused a robust reduction in weight across different ages (*Figure 1*). Moreover, male and female UPS mice spent less time exploring novel objects and showed reduced contextual freezing, findings that were replicated in two independent cohorts (*Figure 2* and *Supplementary file 1*) and in additional studies conducted in the lab (*Figure 2—figure supplement 2*).

## Conclusion

Using high resolution dMRI we show that UPS induces several neuroanatomical changes in the adult rodent brain that mimics those seen in individuals exposed to CM. Some of the most important examples include alterations in global network connectivity, reduced volume and FA in the corpus collosum, frontal cortex atrophy, and increased AMY size, findings that were seen in both males and females. In contrast, exposure to UPS induced sex-specific changes in fronto-limbic connectivity, raising the possibility that these connections may mediate different behavioral outcomes in males and females. This possibility is consistent with findings from a moderated-mediation analysis indicating that UPS-induced alterations in connectivity between the AMY and the vHC is responsible for reduced contextual freezing behavior in males but not in females. These findings reinforce the

continued need for additional research into the moderating effects of sex on CM-induced outcomes in both humans and animals.

# Materials and methods

## Key resources table

| Reagent type (species) or resource | Designation | Source or reference | Identifiers | Additional information |
|---|---|---|---|---|
| Strain, strain background (*Mus musculus*) | Balb/cByj | Jax: 001026 | RRID:MGI:5654246 | |
| Commercial assay or kit | ELISA Corticosterone Kit | Arbor Assays, Ann Arbor, MI | Cat. # K014-H1 | |
| Software, algorithm | DTIStudio | http://www.mristudio.org | Version 3.0.3 | |
| Software, algorithm | MRtrix | http://www.mrtrix.org PMID:26499812 | Version 3.0.2 | |
| Software, algorithm | GRETNA software | GRETNA PMID:26175682 | Version 2.0 | |
| Software, algorithm | SPSS | SPSS, IBM Corp. Armonk, NY | Version 24.0 | |
| Software, algorithm | GraphPad Prism, La Jolla California USA | GraphPad Prism, La Jolla California USA | Version 8.1.0 | |
| Software, algorithm | Matlab | MathWorks, Natick, MA 01760–2098 | Version 2019B | |

## Animals

BALB/cByj mice (Stock # 001026, Jackson Laboratories) were housed in standard Plexiglas cages and kept on a standard 12:12 hr light–dark cycle (lights on at 7:00 AM), with food provided ad libitum and constant temperature and humidity (23 ± 1°C and 43% ± 2). Background noise in the room was kept at dB: 56.5, which was significantly lower compared to the levels recorded (dB: 65.9) in our previous studies (*Johnson et al., 2018*). All studies were approved by the Institutional Animal Care and Use Committee (IACUC) at Yale University, protocol #2020–10981, and were conducted in accordance with the recommendations of the NIH Guide for the Care and the Use of Laboratory Animals.

## Early-life stress models

Thirty females and seven male BALB/cByj mice, 8–10 weeks old, were purchased from the Jackson Laboratory and allowed to acclimate for 12 days in our facility. Breeding cages were set up using a 2:1 female to male harem in standard mouse Plexiglas cages with two cups corncob bedding and no nesting material. Visibly pregnant dams were transferred to 'maternity cages' containing two cups corncob bedding with no nesting material and three chow pellets on the floor. New females were added to the harem cages. On postnatal day (P0) litters were culled to five to eight pups and randomized to either CTL or UPS conditions. Mice raised under CTL conditions were provided with two cups corncob bedding (500cc) and one 5 × 5 cm nestlet per cage. Bedding for CTL condition was changed on P14 and P21. UPS litters were provided with half cup of corncob and no nesting material from P0 to 25 with bedding changes on P7, P14, and 21. In addition to LBN material, UPS litters were separated from their dam for 1 hr on P14, P16, P17, P21, P22, and P25 (*Figure 1A*). During the separation period, the dam was transferred to a new cage followed by the individual transfer of the pups to a different cage containing clean corncob bedding, and the home cage was briefly shaken

to evenly spread the bedding and disrupt the nest. At the end of the 1 hr separation, the pups were individually returned to their home cage followed by the return of the dam. All animals were weighed at P14, P26, and at the time of tissue collection. At the time of weaning (P26) animals were group-housed with same sex littermates with two cups corncob bedding and no nesting material. Cages were changed weekly but were otherwise left undisturbed until behavior testing commenced at P70.

## Behavioral testing

### General strategy

Based on our previous work we estimated that roughly 13 mice per rearing condition and sex are needed to achieve a power of 0.8, using an effect size (Cohn's d = 1.15) and α = 0.05 (two tails) (*Johnson et al., 2018*). Using these a priori power calculations we conducted behavioral testing in a cohort of 12–23 mice per group from four to six litters, for a total of four groups (i.e. CTL-males, UPS-males, CTL-females, and UPS-females; *Figure 2*). To Identify robust behavioral outcomes in lieu of corrections for multiple behavioral testing, we repeated the behavioral work using a second cohort consisting of n = 9–19 mice per group from four to six litters. Behavioral data for cohort 1 and cohort 2 are summarized in *Figure 2—figure supplement 1*. Reproducible outcomes in the novel object exploration and contextual fear conditioning were further confirmed using additional cohorts of animals (*Figure 2—figure supplement 2*).

### Handling

All animals were handled each morning for 3 days prior to the beginning of behavioral testing. Animals were individually removed from the cage and placed onto the back of the hand of the experimenter, and allowed to explore the experimenter hand for 30 s. The animal was placed into a holding cage until all animals from that home cage had been handled, at which time all animals were returned to the home cage. At the end of third day of handling, animals were placed into the behavioral room and allowed to acclimate for 2 hr prior to the onset of behavioral testing. Behavioral testing was conducted between 1230 and 1700 hr daily.

### Exploratory behaviors

Exploratory behavior was tested using the open-field test, the elevated plus maze, and exploration of novel objects. In the open-field test, mice were allowed to explore a 50 × 50 cm arena (lux 60) for 5 min during which the distance traveled and the time spent in the inner 15 cm area was measured using the EthoVision tracking system (Noldus Information Technology). For the elevated plus maze (EPM), the mice were placed in the middle of a standard elevated plus maze (each arm is 10 × 50 cm long; open arm lux 120 and closed arm lux 60) facing an open arm and allowed to explore the maze for 5 min. The time spent exploring the open and closed arms was determined using the EthoVision tracking system. To assess the approach behaviors in a novel context, mice we placed at the right-hand corner of a 50 × 25 cm box (lux 100) with corncob bedding covering the bottom and two identical objects located at the opposite end of the box. Animals were allowed to explore for 5 min, the sessions were recorded, and an experimenter blind to sex and condition scored the amount of time each animal spent in direct contact with the objects.

### Fear-conditioning

On training day, animals are placed into a Med Associates' fear conditioning chamber (Part # VFC-008 with interior dimensions of 29.53 × 23.5×20.96 cm) with a grid floor (Part # ENV-005FPU-M: 29.21 × 29.21×6.05 cm), in the presence of a 2% lemon scent diluted in 70% EtOH. Animals were allowed 300 s to explore the chamber before the onset on tone-shock pairings. Animals were presented with 30, 0.5 s discontinued tones (7500 Hz, 80 dB) that co-terminated with a 1 s 0.65 mA foot shock. Discontinued tones (also known as pips) represent a more ethological relevant acoustic cues (*Kent and Brown, 2012*) as they reliably elicit efficient freezing and have been used as an US in fear conditioning previously (*Kent and Brown, 2012*; *Ito et al., 2009*). This pattern was repeated for a total of five tone-shock pairings with a variable inter-trial interval (30–180 s). There was a 30 s period at the end of the session following the last pairing. Freezing behavior was recorded using Med Associates' Video Freeze software (V2.6.5.81; Med Associates Inc, St. Albans, VT). Animals were then

returned to their home cage until testing. The apparatus was cleaned with 70% EtOH between each session. Twenty-four hours following training, animals were returned to the testing room and placed into the same context (grid floor with a 2% lemon scent diluted in 70% EtOH); freezing behavior was monitored for 10 min and the first 5 min was scored. On the third day of testing, animals were placed into a novel context (walls were replaced with an opaque plastic sheet to round off the walls, a thick mesh netting covered the grid floor, and animals were exposed to a 2% peppermint scent diluted in 70% EtOH). The same five-tone presentation schedule from training was employed in the absence of the shock, and freezing behavior was continually monitored.

## Tissue collection/processing

All tissues were collected between 1030 and 1230 hr to minimize the changes associated with circadian rhythm and was completed 5–6 days following the conclusion of behavioral testing. Briefly, one at a time, animals were anesthetized with chloral hydrate (100 mg/kg) and then placed into a transport cage with some bedding/food from their home cage. Once the animal had achieved sedation (unresponsive to toe-pinch), it was transported to a separate room within the animal facility where a cardiac puncture was performed to collect roughly 100 μl of blood in tubes containing heparin. These were then spun down (5000 × g for 5 min) to collect serum and assess corticosterone levels. Mice were then transcardially perfused using cold PBS/heparin (50 units/ml) solution followed by 10% formalin (polyScience). Adrenal glands were then dissected and weighed. After perfusion, mice were decapitated, and intact skulls were post-fixed for 24 hr at 4°C in 10% formalin, then transferred to sterile 1× PBS (pH 7.4), and left at 4°C until transfer to the imaging facility at NYU for ex vivo dMRI studies (n = 6 mice per condition and sex). These numbers were based on an effect size of (Cohn's d = 2.0), α = 0.05 (two tails), and power = 0.8 (*Johnson et al., 2018*). To further confirm the generalizability of the sex-specific changes we found in fronto-limbic connectivity we expanded the number of mice scanned in each rearing and sex condition to 12–13, for a total of 50 mice (*Figure 10*). Corticosterone levels were determined using ELISA (Cat. # K014-H1, Arbor Assays, Ann Arbor, MI).

## dMRI studies

Upon arrival at the NYU imaging facility, brains were equilibrated with Gadolinium (0.2 mM) in PBS for 1 week at 4°C and scanned by an experimenter blind to the sex and rearing condition. Scanning was conducted overnight using a 7-Tesla MR system equipped with a four-channel cryogenic probe for enhanced sensitivity (*Ratering et al., 2008*). Images were acquired using a modified 3D GRASE sequence (*Wu et al., 2013*) with the following parameters: echo time (TE)/repetition time (TR) = 33/400 ms, 100 μm isotropic resolution, two non-diffusion weighted images ($b_0$s) and 60 diffusion weighted images (DWIs) with a diffusion weighting (b) of 5,000 s/mm$^2$, and a total imaging time of 12 hr. DTIStudio (http://www.mristudio.org) was used to align all DWIs to the average of $b_0$s to remove small sample displacements due to vibrations during the long scan and compute average DWI (aDWI) and FA maps using the diffusion tensor model. The aDWI and FA maps were normalized to an MRI-based atlas (*Chuang et al., 2011*; *Arefin et al., 2019*) using the dual-channel (aDWI+FA) large deformation diffeomorphic metric mapping (LDDMM) (*Ceritoglu et al., 2009*). To assess global and regional AMY connectivity, we transferred 28 structural labels (i.e. 14 nodes and 182 connectomes in each hemisphere) to the subject images using the inverse mapping from LDDMM. The 14 brain regions (aka nodes) chosen for the analysis include the AMY, CB, dHP, dorsal striatum (STRd), lateral hypothalamic zone (LZ), medial hypothalamic zone (MEZ), MO, OB, periventricular zone (PVZ), PFC, somatosensory cortex (SS), thalamus (TH), vHP, and ventral striatum (STRv). These nodes were selected because they show UPS-mediated volumetric and FA alterations using unbiased voxel analyses (*Figures 3* and *4*), are highly connected based on the Allen Mouse Brain Connectivity Atlas (*Oh et al., 2014*), and comply with principles for node selection outlined by *Kaiser, 2011*. These specify that nodes should be (1) non-overlapping, (2) have unique set of connections to other nodes, and (3) be well delineated using standard parcellation scheme that is comparable across species, with later point made possible by the registration of our MRI-based atlas (*Chuang et al., 2011*; *Arefin et al., 2019*). The transferred structural labels were visually examined and showed good agreement with the corresponding structures in the subject images. For each pair of ipsilateral structures, random seeds were first assigned to a structure (e.g. the AMY). Next, the number of seeds

proportional to the volume of the structure and streamlines from the seeds passing through the target structure (e.g. hippocampus) were reconstructed using a probabilistic tractography method implemented in MRtrix (http://www.mrtrix.org) as described previously (*Wu and Zhang, 2016*). The numbers of streamlines between regions were used as a measure of the connection strength. GRETNA software (*Wang et al., 2015*) was used to quantify global and regional properties of the structural connectome using principles of graph theory (*Bullmore and Sporns, 2009*). Global features included measurement of global efficiency (Geff) and the small-worldness (SW) and were normalized with respect to 1000 simulated random networks with equal distribution of edge weight and node strength as reported previously (*Rubinov and Sporns, 2011*; *Schlemm et al., 2017*). Furthermore, local graph parameters, such as nodal clustering coefficient (NCp), nodal efficiency (Neff), and degree centrality (Dcent) (*Watts and Strogatz, 1998*; *Rubinov and Sporns, 2010*), were calculated to capture alterations in the vicinity of the left and right AMY. All imaging scans are available at DOI: https://doi.org/10.35092/yhjc.12367658.

## Moderated-mediation analysis

A moderated-mediation regression analysis was conducted using the *Preacher and Hayes, 2008* bootstrapping procedure (5000 bootstrap resamples) and associated SPSS macro (Model 58) to test the moderating impact of sex on UPS-effects on the number of streamlines between the AMY and vHC and the relationship between these projections and freezing behavior in the contextual fear conditioning test (see *Figure 7A* for visual depiction of path model). For this analysis, rearing condition (dummy-coded; 0 = CTL reared; 1 = UPS reared) was entered into the model as the independent variable (X variable), the number of AMY–vHC streamlines (continuous, mean-centered) was entered into the model as the mediator (M variable), sex (dummy-coded; 0 = male, 1 = female) was specified as the moderator (W variable), and percent freezing was included as the outcome variable (Y variable; log-transformed to correct positive skew, per convention for regression analyses).

## Statistical analysis

Statistical analyses were done using SPSS (IBM Corp. Released 2016. IBM SPSS Statistics for Windows, Version 24.0. Armonk, NY; IBM Corp.) and visualized with GraphPad Prism version 8.1.0 for MacOS (GraphPad Software, La Jolla, California, USA). Animals that were >2 s.d. above or below the mean were eliminated from behavioral analysis. Data were examined using a two-way ANOVA with rearing condition (CTL or UPS) and sex as fixed factors. Behavioral testing was limited to a priori defined and well-accepted endpoints and used p<0.05 (two tails) to identify significant outcomes. A 2 × 2 ANOVA, written in Matlab, was used to assess the effects of UPS, sex, and their interaction on maps of local volumetric changes and FA using voxel-based analysis (*Ashburner and Friston, 2000*). The minimum number of voxels used to define a volumetric cluster and levels for false-base discovery rate correction for multiple comparisons differ depending on whether main effects of rearing, sex, or interaction where tested, and these are specified in the results section. Repeated measures ANOVA was used to assess the effect of laterality as a within-subject variable and sex and rearing as between-subject variables on the following four projections: (1) AMY–PFC, (2) AMY–vHC, (3) AMY–dHC, and (4) vHC–PFC. Significant interactions were followed by post-hoc comparisons using Tukey's HSD or Sidak's test. These same procedures were used to analyze laterality in both global and AMY connectivity. To account for the hierarchical nature of the data, additional linear mixed-effects models were conducted on the key-dependent measures from this experiment to ensure that outcomes were driven by condition or sex and not maternal care or litter per se (*Figure 2—figure supplement 2*).

## Acknowledgements

This work was supported by: NIMH R01MH119164 (AK and JZ), NIMH R01MH118332 (AK and JZ), NCATS TL1 TR001864 (JDW), and R01NS102904 (JZ). The preprint is available on Authorea.

## Additional information

### Funding

| Funder | Grant reference number | Author |
|---|---|---|
| National Institute of Mental Health | R01MH119164 | Jiangyang Zhang<br>Arie Kaffman |
| National Institute of Mental Health | R01MH118332 | Jiangyang Zhang<br>Arie Kaffman |
| National Center for Advancing Translational Sciences | TL1 TR001864 | Jordon D White |
| National Institute of Neurological Disorders and Stroke | R01NS102904 | Jiangyang Zhang |

The funders had no role in study design, data collection and interpretation, or the decision to submit the work for publication.

### Author contributions

Jordon D White, Conceptualization, Data curation, Formal analysis, Supervision, Funding acquisition, Validation, Investigation, Visualization, Methodology, Writing - original draft, Project administration, Writing - review and editing; Tanzil M Arefin, Conceptualization, Data curation, Software, Formal analysis, Supervision, Funding acquisition, Investigation, Methodology, Writing - original draft, Project administration, Writing - review and editing; Alexa Pugliese, Data curation, Software, Formal analysis, Investigation, Methodology, Writing - review and editing; Choong H Lee, Data curation, Investigation; Jeff Gassen, Software, Formal analysis, Investigation, Writing - review and editing; Jiangyang Zhang, Conceptualization, Data curation, Software, Formal analysis, Supervision, Funding acquisition, Investigation, Methodology, Project administration, Writing - review and editing; Arie Kaffman, Conceptualization, Data curation, Software, Formal analysis, Supervision, Funding acquisition, Validation, Investigation, Visualization, Methodology, Writing - original draft, Project administration, Writing - review and editing

### Author ORCIDs

Arie Kaffman https://orcid.org/0000-0002-7028-8869

### Ethics

Animal experimentation: All studies were approved by the Institutional Animal Care and Use Committee (IACUC) at Yale University, protocol #2020-10981, and were conducted in accordance with the recommendations of the NIH Guide for the Care and the Use of Laboratory Animals.

### Decision letter and Author response

Decision letter https://doi.org/10.7554/eLife.58301.sa1
Author response https://doi.org/10.7554/eLife.58301.sa2

## Additional files

### Supplementary files

• Source data 1. Behavioral data cohort 2.

• Supplementary file 1. Summary of behavioral result in cohort 2. CA – closed arms, EPM – elevated plus maze, OA – open arms, OF – open field test, s – seconds. Significant outcomes are shown in red font.

• Supplementary file 2. Linear mixed-effect model (LMEM) for key outcomes in unpredictable postnatal stress (UPS) mice. To examine whether the reported effects were attributable to *litter effect*, we conducted linear mixed-effects model using R version 3.6.0 (*R Development Core Team, 2019*) and the following packages: lme4 version 1.1–21 (*Bates et al., 2015*) and lmerTest version 3.1–0

(*Kuznetsova et al., 2017*), with litter included in each model as a random effect. Importantly, these models provided a second test of key behavioral endpoints while controlling for the hierarchical structure of the data as animals were nested within litters. Results again revealed a main effect of rearing condition on contextual fear conditioning with reduced freezing found in UPS- compared to CTL-reared animals, b = −17.78, SE = 6.22, t = −2.86, p=0.02. Similarly, time spent exploring novel objects was reduced in UPS compared to CTL animals, b = −4.81, SE = 2.13, t = −2.25, p=0.053. There was also a main effect of sex in NOR, with females demonstrating greater exploration than males, b = 3.87, SE = 1.51, t = 2.57, p=0.01. The results of the tractography linear mixed-effects model were also consistent with the reported ANOVA, revealing a significant interaction between sex and rearing condition on streamlines between the amygdala and ventral hippocampus in the left hemisphere, b = −3851.21, SE = 901.54, t = −4.27, p=0.0004. Together, the results of these follow-up models support those reported in the main text and suggest that litter (or maternal behavior) had minimal effect on the key dependent measures. R codes and output for the linear mixed model analysis are available in *Source data 1*.

- Supplementary file 3. Summary table of local volumetric changes induced by unpredictable postnatal stress using voxel-based morphometric analysis with minimal cluster size >25 voxels, FDR < 0.1, p<0.0105. n = 12 mice per condition.

- Transparent reporting form

## Data availability

All imaging data are deposited at https://doi.org/10.35092/yhjc.12367658.

The following dataset was generated:

| Author(s) | Year | Dataset title | Dataset URL | Database and Identifier |
|---|---|---|---|---|
| White JD, Arefin TM, Pugliese A, Lee CH, Gassen J, Zhang J, Kaffman A | 2020 | Early Life Stress Mouse Brain MRI Dataset | https://doi.org/10.35092/yhjc.12367658 | figshare, 10.35092/yhjc.12367658 |

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
