## [Decision Letter]

**Acceptance summary:**

The authors explored how sex mediates the effect of early life stress on brain and behaviour in mice. They found that such stress increases the connectivity in males whereas it had no or even an opposite effect in females; this change in connectivity in turn influenced how males, but not females, behaviourally responded to stressful situations. This paper thus provides important biological insights into the differential effects of stress on the male and female brain.

**Decision letter after peer review:**

Thank you for submitting your article "Early life stress causes sex-specific changes in adult fronto-limbic connectivity" for consideration by *eLife*. Your article has been reviewed by three peer reviewers, one of whom is a member of our Board of Reviewing Editors, and the evaluation has been overseen by Timothy Behrens as the Senior Editor. The reviewers have opted to remain anonymous.

The reviewers have discussed the reviews with one another and the Reviewing Editor has drafted this decision to help you prepare a revised submission.

Summary:

This manuscript uses behavioural testing and diffusion MRI to assess the effects of early life stress on brain and behaviour. The authors employed a modified version of the limited bedding and nesting protocol developed by Baram, which included maternal separation and nest disruption (ranging from P0-P25)- which the authors term unpredictable postnatal stress (UPS). In the current studies, the authors tested control and UPS mice on the elevated plus maze (EPM), open field (OF), novel object approach, as measures of anxiety-like behavior. Interestingly, the authors failed to replicate their initial findings of elevations in anxiety-like behavior in male mice. The authors instead found no main effect of treatment or interaction with sex for OF. The authors found an anxiolytic effects of UPS on the EPM test, with UPS mice spending more time in the open arms, with no effect of sex or interaction. Finally, the authors found an anxiogenic effect of UPS rearing on novel object approach, with a significant effects of sex that did not replicate in an internal replication study. The authors observe very interesting effects of their manipulation on regional brain volume (increase in some regions, with reductions in others, that appear to be sex specific). The authors also note significant effects of UPS rearing on regional connectivity by DTI, with significant effects on connectivity between limbic, hippocampal, and prefrontal structures. The authors try to relate individual differences in strength of connections (based upon tractography) with levels of freezing behavior. Curiously, for this measure, the author observed significant effects of sex and rearing condition on amygdala-hippocampal streamlines that appear to be predictive of levels of freezing behavior in males but not females.

Essential revisions:

This is a timely paper addressing an interesting topic. The experiments are well designed, employ a number of approaches, along with technical rigor. However, a number of questions arise related to failure to replicate prior effects from the lab, as well as the ability to replicate main effects between studies within this report.

The authors use their previously published UPS model, which is an adaptation of the more commonly used limited bedding model. Using their UPS model, the authors were not able to replicate their previous findings of UPS effects on anxiety-like behavioral phenotypes, which evokes the question of how robust this model is compared to more commonly used models of ELS. The authors discuss the possible impact of increased noise in the room during testing for anxiety-like behavior tasks and the potential sensitivity of these assays to environment and experimenter which may have driven the failure to replicate. However, the authors also appeared to have difficulty in replicating some of the more robust assays (some measures of fear learning), possibly indicating that the variability may be at the level of efficacy of stress induction and not necessarily and solely variability at the time of phenotypic testing. Perhaps the authors could comment on how replicable the neural connectivity findings in the current study are expected to be in future attempts to replicate this work. Ideally, an additional replication cohort or more robust comparison of existing literature data would provide stronger evidence about which effects of UPS and/or ELS in general are robust.

The n=6 per group for MRI is too low; most power analyses indicate that around 10-12 mice per group are more appropriate (i.e. that's where you hit the point of diminishing returns for adding more mice), and should be even higher if interactions are tested. Please expand the sample to more reasonable numbers. The fact that the authors did not find many of the robust sexually dimorphic brain regions (i.e. BNST was found at q<0.3, but MeA and MPOA, etc., appear to be missing in Figure 3—figure supplement 1) further adds to the doubt that the MRI study as is was adequately powered.

Please explain why were Balb/c mice used. They tend to have very poor maternal care compared to, for example, Bl6 or CD1, and most literature on ELS has been on Bl6 mice.

How were multiple comparisons corrected or accounted for in the analysis of the behaviour data? If not at all, as it seems, this needs to be remedied or extensively justified.

How accurate was the tract tracing? The streamlines shown in Figure 5 do not look all that much like the tract tracing from the Allen institute's connectivity atlas – can the authors overlay those maps or find an alternate way to measure their tracing accuracy? It is hard to interpret different streamlines by group or sex when it is not clear whether diffusion based tracing truly reflects underlying wiring.

The Introduction needs to be expanded. For example, the authors highlight recent reviews that speculate about the potential importance of the type of stressor and impact of different forms of early life stress on neurodevelopmental outcomes (CM falling along multiple dimensions), along with sex as a moderating factor (Mclaughlin and Sheridan). I would like to point the authors to Demaestri et al., 2020 and Bath, 2020, further bolster these points and supplement the excellent work done by this group in their 2018 Translational Psychiatry paper. Furthermore, while the authors do acknowledge some previous work on sex effects and childhood maltreatment in the Introduction and Discussion of their manuscript, the only paper that is cited is the authors own review by White and Kaffman. The background literature on sex effects should be discussed with reference to a broader body of appropriate literature.

[Editors' note: further revisions were suggested prior to acceptance, as described below.]

Thank you for resubmitting your article "Early life stress causes sex-specific changes in adult fronto-limbic connectivity that differentially drive learning" for consideration by *eLife*. Your revised article has been re-reviewed by the Reviewing Editor and Timothy Behrens as the Senior Editor.

The reviewers have discussed the reviews with one another and the Reviewing Editor has drafted this decision to help you prepare a revised submission.

We believe that this revision has addressed the great majority of the concerns raised in the first round. The only remaining concern relates to our disappointment that the authors opted not to increase the sample size from the too-small n=6 for the imaging study. The authors did add a disclaimer in their discussion about their limited power. We ask that they move the disclaimer to earlier in the discussion, make explicit in the Discussion that the n=6 was designed to replicate an initial hypothesis, and that they were underpowered for the types of brain wide sex by condition interactions that were carried out.

---

## [Author Response]

Summary:This manuscript uses behavioural testing and diffusion MRI to assess the effects of early life stress on brain and behaviour. The authors employed a modified version of the limited bedding and nesting protocol developed by Baram, which included maternal separation and nest disruption (ranging from P0-P25)- which the authors term unpredictable postnatal stress (UPS). In the current studies, the authors tested control and UPS mice on the elevated plus maze (EPM), open field (OF), novel object approach, as measures of anxiety-like behavior. Interestingly, the authors failed to replicate their initial findings of elevations in anxiety-like behavior in male mice. The authors instead found no main effect of treatment or interaction with sex for OF. The authors found an anxiolytic effects of UPS on the EPM test, with UPS mice spending more time in the open arms, with no effect of sex or interaction. Finally, the authors found an anxiogenic effect of UPS rearing on novel object approach, with a significant effects of sex that did not replicate in an internal replication study.

While we do acknowledge that the anxiety findings reported in Johnson et al., 2018 are not reproduced in this paper, we felt that there were some important details from this summary statement that needed corrections and clarifications. First, in both cohorts from this paper (and others tested since) there are no clear indications of an anxiety phenotype in OF or EPM, as non-significant condition effects were replicated in both OF cohorts, and only 1 cohort found significant condition effects in the open arm of the EPM (an effect that hinted at an anxiolytic effect of UPS, Figure 2C). The EPM finding is important as it highlights the difficulties of replicating outcomes with this test even under identical conditions. We believe that environmental differences that were present between the animal facility utilized in Johnson et al., 2018 and the room in current use likely interacted with the stress paradigm in an unforeseen way to intensify the nature and severity of the original ELS-paradigm. Second, the anxiogenic effect of rearing in the novel object exploration task was replicated across the two cohorts and in additional cohorts. Third, the behavioral summary did not mention that UPS reduced freezing behavior in the contextual fear conditioning (Figure 2F), an effect that was also replicated across the two cohorts and in additional animals tested. To further demonstrate the robustness of outcomes in the novel object exploration and CFC, we included data from additional groups of animals that replicated these outcomes. This information is available in Figure 2—figure supplement 2 and was added to the revised Results section by noting that: “To further demonstrate the robustness of the rearing effect on outcomes in the novel exploration test (Figure 2E) and contextual fear condition (Figure 2F) we replicated these findings in additional cohorts of mice (Figure 2—figure supplement 2) and in the Discussion:” Our inability to replicate behavioral outcomes in the EPM in the first and second cohorts underscores the challenges of using this test even under practically identical conditions (Supplementary file 1) and highlights the need to develop more robust behavioral tests of anxiety, see (Oberrauch et al., 2019, Meyer et al., 2019, Prevot et al., 2019) for some encouraging examples. Finally, we note that: “Consistent with our previous work (Johnson et al., 2018) we found that UPS caused a robust reduction in weight across different ages (Figure 1). Moreover, male and female UPS mice spent less time exploring novel objects and showed reduced contextual freezing, findings that were replicated in 2 independent cohorts (Figure 2 and Supplementary file 1) and in additional studies conducted in the lab (Figure 2—figure supplement 2).”

The authors observe very interesting effects of their manipulation on regional brain volume (increase in some regions, with reductions in others, that appear to be sex specific). The authors also note significant effects of UPS rearing on regional connectivity by DTI, with significant effects on connectivity between limbic, hippocampal, and prefrontal structures. The authors try to relate individual differences in strength of connections (based upon tractography) with levels of freezing behavior. Curiously, for this measure, the author observed significant effects of sex and rearing condition on amygdala-hippocampal streamlines that appear to be predictive of levels of freezing behavior in males but not females.

We would like to clarify that we also found significant effects of rearing on volumetric changes (Figure 3), alterations in FA (Figure 4), and global connectivity (Figure 8). These outcomes were similarly affected in males and females and replicated some of the most robust imaging findings in humans. In contrast, males and females showed very different patterns, with a significant interaction, when fronto-limbic connectivity was assessed. This is an important finding that highlights the need to consider both global and specific circuits when addressing the question of how sex moderates the outcomes of early adversity. This point has not received much attention in the literature to date and is nicely highlighted in the paper. We emphasized the significance of this novel finding in the Results section by writing: ”Together, these findings highlight the exquisite specificity by which UPS alters structural connectivity in male and female mice” and noted in the revised Discussion that: “The role that sex plays in CM-associated outcomes is a growing area of interest (White and Kaffman, 2019, Bath, 2020), and it is becoming increasingly clear that the influence of sex may only be seen in certain behaviors, neurodevelopmental processes, and/or specific circuits (White and Kaffman, 2019, Bath, 2020, Demaestri et al., 2020). This is in line with findings reported here. For example, here we report similar impact of UPS on body weight, object exploration, contextual conditioning, global connectivity and local volumetric and FA changes in males and females. In contrast, there are robust moderating effects of sex on UPS-induced outcomes in baseline corticosterone levels, fronto-limbic connectivity, and amygdala nodal efficiency.”

Essential revisions:This is a timely paper addressing an interesting topic. The experiments are well designed, employ a number of approaches, along with technical rigor. However, a number of questions arise related to failure to replicate prior effects from the lab, as well as the ability to replicate main effects between studies within this report.

We appreciate the kind words and have made a significant effort to address all the comments below. We disagree however with the assertion that main effects were not replicated within the current study (see also response to statement 1 above). For example, significant main effects of condition were replicated in the 2^nd^ cohort in the novel object exploration task and the contextual conditioning tasks, and a non-significant main effect of rearing were replicated in the both OF measures, exploration of the closed arm of the EPM, baseline freezing behavior, freezing behavior over the course of the cued-training paradigm, and freezing during cue-induced recall. Overall, there is only 1 instance (out of 9 possible) when a main effect of rearing is not replicated with the second cohort. Further, a significant main effect of sex is replicated in freezing during cue-induced recall, and non-significant main effects of sex are replicated for OF and EPM measures, baseline freezing behavior, freezing behavior over the course of the cued-training paradigm, and contextual freezing. We feel that this internal replication conveys the robustness of the behavioral phenotype described in this study.

The authors use their previously published UPS model, which is an adaptation of the more commonly used limited bedding model. Using their UPS model, the authors were not able to replicate their previous findings of UPS effects on anxiety-like behavioral phenotypes, which evokes the question of how robust this model is compared to more commonly used models of ELS. The authors discuss the possible impact of increased noise in the room during testing for anxiety-like behavior tasks and the potential sensitivity of these assays to environment and experimenter which may have driven the failure to replicate. However, the authors also appeared to have difficulty in replicating some of the more robust assays (some measures of fear learning), possibly indicating that the variability may be at the level of efficacy of stress induction and not necessarily and solely variability at the time of phenotypic testing.

We direct the reviewers to response 1 above and believe that difficulty in reproducing behavioral effects is not unique to the UPS procedure, and has been reported by other commonly used ELS paradigms such as LBN, maternal separation and handling (Wang et al., 2020, White and Kaffman, 2019, Walker et al., 2017, Loi et al., 2015, Tractenberg et al., 2016, Murthy and Gould, 2018). This is not surprising given the large number of variables that affect ELS outcome (e.g. differences in travel conditions and composition of animals delivered from Jackson labs, housing conditions, maternal behavior, details related to restricting bedding, separation procedure and nest disruption) and the additional variables related to administering the OF and EPM (e.g. light, noise, smell, sex of the researcher). It is difficult to know what factors drive the difficulties of replicating anxiolytic findings in rodents, but this is an important issue that needs to be reported and addressed. To address this issue, we utilized internal replication cohorts and then focused on the behavioral outcomes that were consistent across those cohorts. In our case, reduced freezing in response to the training context and reduced exploration of novel objects were robust (Figure 2, Figure 2—figure supplementary 2, Supplementary file 1). In addition, the use of long-term automated behavioral assessment in the home cage appear to be a promising approach for identifying robust behavioral phenotypes, as demonstrated by (Goodwill et al., 2018), and (Prevot et al., 2019). Finally, learning paradigms of ambiguous cues will likely provide more robust outcomes (Oberrauch et al., 2019, Meyer et al., 2019) when compared to brief measurements of innate defensive behavior.

Perhaps the authors could comment on how replicable the neural connectivity findings in the current study are expected to be in future attempts to replicate this work. Ideally, an additional replication cohort or more robust comparison of existing literature data would provide stronger evidence about which effects of UPS and/or ELS in general are robust.

We addressed this important point in several ways. First, we confirmed the generalizability of our sex-specific tractography findings by increasing the number of mice scanned to 12-13 per rearing and sex condition for a total of 50 mice (Figures 10-12). Second, we note that our dMRI connectivity data have replicated our previous rsfMRI findings of increased connectivity between the 1) amygdala and the ventral hippocampus, 2) amygdala and dorsal hippocampus, and 3) amygdala and the prefrontal cortex in adult UPS male mice (Johnson et al., 2018). Third, we found the exact same pattern when assessing the left and right hemispheres using both rsfMRI and dMRI (Figure 5—figure supplement 2 and 3 and Figure 6—figure supplement 1). This is especially important for the dMRI findings reported here because the analysis of left hemisphere was done about two months after conducting the tractography for the right hemisphere. Fourth, similar increase in connectivity between the amygdala and the PFC was reported in rats exposed to LBN (Bolton et al., 2018). To address this issue, we added a new section at the end of the Results section titled: Generalizability of the sex-specific tractography findings, which demonstrates the robustness of these findings in a large group of animals.

We revisit this concern in the Discussion by stating: ”Our dMRI tractography shows good agreement with anterograde viral tracing reported by the Allen Mouse Brain Connectivity Atlas (Figure 5—figure supplement 1), suggesting that the tractography data shown here represents true anatomical projections. Using this approach we found a robust increase in fronto-limbic connectivity in males exposed to UPS (Figures 5-6).These structural changes are consistent with our previous resting-state fMRI findings in UPS males (Johnson et al., 2018) and those reported in male rats exposed to LBN (Bolton et al., 2018). Similar patterns were seen in both the left and the right hemispheres (Figure 5—figure supplement 1 and 2, and Figure 6—figure supplement 1) and were further confirmed using a larger cohort of mice (Figure 10).”

The n=6 per group for MRI is too low; most power analyses indicate that around 10-12 mice per group are more appropriate (i.e. that's where you hit the point of diminishing returns for adding more mice), and should be even higher if interactions are tested. Please expand the sample to more reasonable numbers. The fact that the authors did not find many of the robust sexually dimorphic brain regions (i.e. BNST was found at q<0.3, but MeA and MPOA, etc., appear to be missing in Figure 3—figure supplement 1) further adds to the doubt that the MRI study as is was adequately powered.

The primary goal of this work was to replicate our previous rsfMRI findings of increased fronto-limbic connectivity in male UPS mice. Our previously reported effect size of Cohn’s d = 2.0 was therefore used for sample size calculations. Using this sample size, we were able to replicate our previous functional connectivity data, and to identify multiple changes in volumetric (Figure 3) and FA measurements that were affected by rearing at a FDR < 0.1 (Figure 4). Importantly, many of these changes are consistent with robust changes seen in humans. We consider these to be key findings and emphasized these points in the revised discussion. We agree that this sample size is not likely to have sufficient power to adequately identify sexually dimorphic changes and interactions after rigorous correction for multiple testing. The relatively lower resolution of diffusion MRI compared to T1/T2-weighted MRI and differences in tissue contrasts may also limited our ability to identify additional structures. We have acknowledged these limitations in the Discussion: “Our sample size was not sufficiently powered to identify sexually dimorphic changes after rigorous correction for multiple comparisons, and the relatively lower resolution of dMRI compared to T1/T2-weighted MRI and differences in tissue contrasts may have also limited our ability to capture these differences. Nevertheless, we have identified several of the previously known sexually dimorphic brain regions including, the BNST and medial preoptic area in the hypothalamus, hippocampus, thalamus, olfactory bulb, striatum, and the sensory cortex (Figure 3—figure supplement 1). These findings are consistent with a previous study in C57BL/6 mice (Qiu et al., 2018) and in humans (Gillies and McArthur, 2010, Ruigrok et al., 2014, Goldstein et al., 2001), underscoring the conserved nature of these sexual dimorphic alterations.”

Please explain why were Balb/c mice used. They tend to have very poor maternal care compared to, for example, Bl6 or CD1, and most literature on ELS has been on Bl6 mice.

Balb/cByj mice were used in this study in order to build upon the extensive studies from the lab using this strain (Wei et al., 2010, Wei et al., 2012, Wei et al., 2014, Wei et al., 2015, Delpech et al., 2016) and to specifically replicate and extend findings in this strain (Johnson et al., 2018). We have previously characterized maternal behavior in Balb/cByj and Bl6 dams under the parameters used in the present experiments (Wei et al., 2010). For example, in the absence of nesting material on P0, B6 dams failed to care for their pups leading to high levels of pup mortality that was not seen in Balb/cByj (Wei et al., 2010). Further, Balb/cByj mice are more sensitive than C57 to the effects of stress including ELS making them an excellent strain to study this issue. Finally, exposure to UPS in this strain replicates multiple imaging findings reported in humans, further supporting the use of this strain as a rodent model of ELS. We addressed these issues in the revised manuscript in the Introduction: “Balb/cByj mice were used due to their high sensitivity to stress, including ELS (Caldji et al., 2004, Zaharia et al., 1996, Francis et al., 2003, Carola et al., 2004, Tractenberg et al., 2016, Wei et al., 2010, Wei et al., 2012, McWhirt et al., 2019, Malki et al., 2015, Flint and Tinkle, 2001, Wei et al., 2014, Wei et al., 2015, Delpech et al., 2016) and because Balb/cByj dams reliably maintain their litters in the complete absence of nesting material (Wei et al., 2010). Further, we have previously shown that UPS produces an elevated anxiety phenotype in both adolescent and adult Balb/cByj offspring (Johnson et al., 2018). Using rsfMRI we found increased fronto-limbic connectivity in UPS male mice that included increased amygdala-prefrontal cortex and amygdala-hippocampus connectivity, the strength of which was highly correlated with anxiety-like behaviors (Johnson et al., 2018). Interestingly, females did not show an elevated anxiety phenotype, suggesting that UPS may affect males and females differently. This original study did not include females in rs-fMRI analysis and is thus unable to speak to potential ELS by sex interactions on fronto-limbic connectivity patterns”. We review the agreement with the human literature in the discussion and in the conclusion sections.

How were multiple comparisons corrected or accounted for in the analysis of the behaviour data? If not at all, as it seems, this needs to be remedied or extensively justified.

Correction for multiple testing is not commonly done in behavioral work. In fact, we are not aware of a single ELS manuscript that has corrected for multiple testing, including several high-profile manuscripts (Goodwill et al., 2018, Bolton et al., 2018, Pena et al., 2017, Honeycutt et al., 2020). The reasons for this are that the effect sizes of many behavioral tests are moderate, at best, and would require extremely large cohorts of animals to accommodate for multiple testing. Moreover, given the correlated nature of many behavioral endpoints (especially those related to anxiety), multiple testing corrections would be too conservative. We feel that it is both more practical and useful to replicate the effects in independent cohorts in order to show the robustness of the phenotype, a tactic that we utilized in the present manuscript (see also responses to summary statement #1 and comment 2). Further, behavioral testing was limited to a priori defined and well accepted endpoints, with p< 0.05 (two tails) used to identify significant outcomes. Finally, all post-hoc comparisons following significant interaction were corrected for multiple comparisons using Tukey’s HSD or Sidak’s tests. We have clarified this in the Materials and methods section by stating that: “To Identify robust behavioral outcomes in lieu of corrections for multiple behavioral testing, we repeated the behavioral work using a second cohort consisting of n = 9-19 mice per group from 4-6 litters. Behavioral data for cohort 1, and cohort 2 are summarized in Supplementary file 1. Reproducible outcomes in the novel object exploration and contextual fear conditioning were further confirmed using additional cohorts of animals (Figure 2—figure supplement 2).” We note that: ”Behavioral testing was limited to a priori defined and well accepted endpoints and used p< 0.05 (two tails) to identify significant outcomes”, “The minimum number of voxels used to define a volumetric cluster and levels for false-base discovery rate correction for multiple comparisons differ depending on whether main effects of rearing, sex or interaction where tested, and these are specified in the Results section”, “Significant interactions were followed by post-hoc comparisons using Tukey’s HSD or Sidak’s test.”

How accurate was the tract tracing? The streamlines shown in Figure 5 do not look all that much like the tract tracing from the Allen institute's connectivity atlas – can the authors overlay those maps or find an alternate way to measure their tracing accuracy? It is hard to interpret different streamlines by group or sex when it is not clear whether diffusion based tracing truly reflects underlying wiring.

We thank you for this suggestion, as we agree it would increase the reader’s confidence in the findings discussed in the current document. Several observations indicate that our tractography data are highly accurate and represent true anatomical differences, most of which have been detailed in response #3 above. Moreover. In Figure 5—figure supplement 1 we now show that there is a good agreement between our streamline projection and the expected anterograde connectivity available at the Allen Mouse Brain Connectivity Atlas (AMBCA). We discuss this in the revised Results section by writing that: “To address this issue, we first confirmed that there is a good agreement between our dMRI projections and the expected anterograde connectivity available from the Allen Mouse Brain Connectivity Atlas (Figure 5—figure supplement 1).” In Figure 5—figure supplement 1 legend we note that full agreement is not expected because dMRI examines tracks that are bi-directional vs. anterograde tracing seen with AAV-GFP (Maier-Hein et al., 2017, Pallast et al., 2020). In addition, the AMBCA tracing was done with AAV1 (Oh et al., 2014) which has some retrograde transport component (Murlidharan et al., 2014), further complicating the comparison with dMRI.

The Introduction needs to be expanded. For example, the authors highlight recent reviews that speculate about the potential importance of the type of stressor and impact of different forms of early life stress on neurodevelopmental outcomes (CM falling along multiple dimensions), along with sex as a moderating factor (Mclaughlin and Sheridan). I would like to point the authors to Demaestri et al., 2020 and Bath, 2020, further bolster these points and supplement the excellent work done by this group in their 2018 Translational Psychiatry paper. Furthermore, while the authors do acknowledge some previous work on sex effects and childhood maltreatment in the Introduction and Discussion of their manuscript, the only paper that is cited is the authors own review by White and Kaffman. The background literature on sex effects should be discussed with reference to a broader body of appropriate literature.

We thank you for this comment and have incorporated these additional references in the Introduction and Discussion lines.

[Editors' note: further revisions were suggested prior to acceptance, as described below.]

We believe that this revision has addressed the great majority of the concerns raised in the first round. The only remaining concern relates to our disappointment that the authors opted not to increase the sample size from the too-small n=6 for the imaging study. The authors did add a disclaimer in their discussion about their limited power. We ask that they move the disclaimer to earlier in the discussion, make explicit in the Discussion that the n=6 was designed to replicate an initial hypothesis, and that they were underpowered for the types of brain wide sex by condition interactions that were carried out.

We respectfully disagree with the statement that “the authors opted not to increase the sample size from the too-small n=6 for the imaging study” and note that we doubled the sample size for all the tractography work (Figures 10, 11, 12) and that this larger cohort further replicated our initial findings. We agree that our initial sample may not have been sufficiently powered to detect rearing by sex interactions for whole brain volumetric and FA voxel analyses and clarified this early in the Discussion by stating that: “This is in line with findings reported here. For example, here we report similar impact of UPS on body weight, object exploration, contextual conditioning, and global connectivity in males and females. Male and female UPS mice also showed similar changes in local volumetric and FA, but our sample size for these measures (n=6) was not sufficiently powered to detect significant sex by rearing interactions after correcting for multiple testing. Nevertheless, this sample size revealed robust moderating effects of sex on UPS-induced outcomes in fronto-limbic connectivity and amygdala nodal efficiency, findings that were further confirmed using a larger cohort of mice.” It is worth pointing out that this sample size (n=6 per group or a total sample of 24 mice) was sufficient to detect several rearing effects that replicated work in humans. Finally, we left the Discussion regarding sample size and power for detecting sex differences unchanged.